# MedCoG: Maximizing LLM Inference Density in Medical Reasoning via Meta-Cognitive Regulation

Yu Zhao[† 1 2]  Hao Guan[2]  Yongcheng Jing[2]  Ying Zhang[1]  Dacheng Tao[2]

## Abstract

Large Language Models (LLMs) have shown strong potential in complex medical reasoning yet face diminishing gains under inference scaling laws. While existing studies augment LLMs with various knowledge types, it remains unclear how effectively the additional costs translate into accuracy. In this paper, we explore how meta-cognition of LLMs, i.e., their self-assessment of their own cognitive states, can regulate the reasoning process. Specifically, we propose MEDCOG, a **Med**ical **Me**ta-**Cog**nition Agent with Knowledge **G**raph, where the meta-cognitive assessments of task complexity, familiarity, and knowledge density dynamically regulate utilization of procedural, episodic, and factual knowledge. The LLM-centric on-demand reasoning aims to mitigate the diminishing returns under scaling law by (1) reducing costs via avoiding indiscriminate scaling, (2) improving accuracy via filtering out distractive knowledge. To validate this, we empirically characterize the scaling curve and introduce *inference density* to quantify inference efficiency. Experiments demonstrate the effectiveness and efficiency of MEDCOG on five hard sets of medical benchmarks, yielding 6.2x inference density. Furthermore, the Oracle study highlights the significant potential of meta-cognitive regulation.

## 1. Introduction

Medical reasoning (Jin et al., 2021; Pal et al., 2022; Jin et al., 2019; Wang et al., 2024a; Hendrycks et al., 2020; Zuo

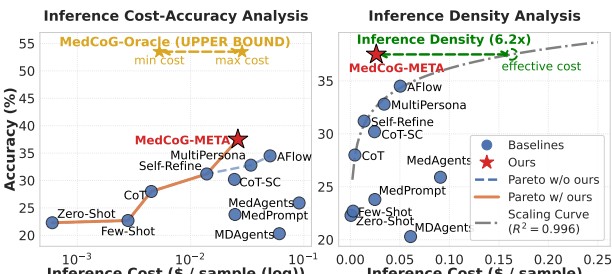

*Figure 1.* (a) Inference Cost-Accuracy Analysis based on GPT-4o on 5 hard sets. (b) Inference Density Analysis. The inference scaling curve is fitted with $R^2 = 0.996$. MEDCOG-META advances the Pareto Frontier and shows $6.2\times$ inference density. MEDCOG-ORACLE reveals the upper bound of meta-cognition regulation.

et al., 2025) represents one of the most challenging frontiers for Large Language Models (LLMs), demanding not only broad biomedical knowledge incorporation but also precise multi-hop reasoning. Early approaches (Nori et al., 2023b;a) treated LLMs as standalone solvers. More recently, the field has shifted towards agentic medical reasoning (Tang et al., 2025; Zhu et al., 2025) that augment models with multi-role playing (Tang et al., 2024; Kim et al., 2024; Wu et al., 2025c), retrieval systems (Wu et al., 2025b; Rezaei et al., 2025; Liang et al., 2025; Xu et al., 2025b), iterative refinement loops (Liu et al., 2025; Tang et al., 2025; Liang et al., 2025), clinical memory (Chen et al., 2025; Lu et al., 2025), etc. While these agents emulate clinical workflows and improve diagnostic reasoning, their growing complexity introduces a critical efficiency bottleneck.

We analyzed the cost-accuracy curves and identified the Pareto frontier of existing approaches ("Pareto w/o ours" in Figure 1 (a)). Subsequently, we fitted the inference scaling curve to this Pareto frontier (Figure 1 (b)). Most approaches align with the logarithmic inference scaling law, i.e., increased computational cost yields progressively smaller performance gains. The additional costs are usually brought by the test-time scaling techniques (Wang et al., 2022; Tang et al., 2024; Kim et al., 2024; Madaan et al., 2023; Zhang et al., 2025a). Given the growing capabilities of foundation LLMs, it remains unclear how effectively the additional costs translate into medical reasoning capability gains.

[†] This work was completed during Yu Zhao's research attachment at NTU. Project Page: https://OreOZhao.github.io/MedCoG-webpage. [1]College of Computer Science, VCIP, DISSec Center, Nankai University, China 300350 [2]Generative AI Lab, College of Computing and Data Science, Nanyang Technological University, Singapore 639798. Correspondence to: Dacheng Tao <dacheng.tao@gmail.com>.

*Proceedings of the 43rd International Conference on Machine Learning*, Seoul, South Korea. PMLR 306, 2026. Copyright 2026 by the author(s).

We first conduct a pilot analysis using reasoning strategies widely adopted in medical and agentic frameworks, including Structural Chain-of-Thought (SCoT) (Wei et al., 2022; Chen et al., 2024a; Wu et al., 2025a), Knowledge Graph retrieval (KG) (Wu et al., 2025b; Chandak et al., 2023), and Memory (Liu et al., 2025; Hu et al., 2025; Zhang et al., 2025b) with historical reasoning experiences.

**(1) Negative Impacts to LLMs:** Contrary to the intuition that more is better, our pilot analysis in Table 1 reveals that augmenting LLMs with KG retrieval or memory does not consistently improve performance, probably attributed to knowledge noise and over-reasoning. This suggests indiscriminate knowledge augmentation can lead to negative impacts on foundation LLMs with already substantial internal capabilities.

**(2) High Potential of Oracle Strategy:** Our Oracle study reveals that LLMs adaptively combining the optimal strategy within the fixed pool achieve a high upper knowledge boundary (Table 1) with low cost (Figure 1 (a)). The observations underscore that the bottleneck of current methods is not the knowledge scope, where 98.98 even surpasses the most advanced LLM (Vals AI, 2026), but rather a lack of a precise allocation mechanism to select strategies adaptively.

To address the bottleneck, we draw inspiration from meta-cognition science. As the ancient adage "Know Thyself" suggests, an intelligent agent should be capable of assessing their own knowledge state and select the corresponding reasoning strategies. By exploiting the meta-cognition of LLMs, we can transition from blindly scaling to LLM-centric on-demand reasoning and enable the self-regulation of different knowledge strategies for LLMs. It allows the LLM to approximate the ideal Oracle performance, thereby mitigating the diminishing returns under inference scaling law via: (1) reducing costs via avoiding indiscriminate scaling, (2) improving accuracy via filtering out distractive knowledge.

To this end, we introduce MEDCOG, a **Med**ical **Me**ta-**Cog**nitive Agent with Knowledge **G**raph where LLM dynamically regulates the reasoning process based on its self-assessments. Specifically, MEDCOG consists of a Meta-Cognition Regulator and an Executor. Inspired by Meta-Cognition science (Schraw, 1998; Tulving et al., 1972), the Regulator governs the Executor through *Monitoring*, *Planning*, and *Evaluating*. We also categorize the knowledge types of the Executor as: *Procedural Knowledge* from SCoT, *Factual Knowledge* from knowledge graph verification, and *Episodic Knowledge* from reasoning experiences. For each sample, the Regulator first monitors the three meta-cognitive dimensions: *Complexity*, *Familiarity*, and *Knowledge Density*, then plans reasoning strategies to activate corresponding knowledge. After the execution is completed, the Regulator further evaluates whether to terminate reasoning.

*Table 1.* The performance of GPT-4o with different reasoning strategies. MEDCOG-ORACLE denotes perfect strategy selection inside our framework. MEDQA-F denotes full set of MedQA (Jin et al., 2021), while -H denotes hard subset (Tang et al., 2025).

| STRATEGY | MEDQA-F | MEDQA-H |
|---|---|---|
| SOTA | O1/96.52 | O3-MINI/53.0 |
| *GPT-4o based* | | |
| ZERO-SHOT | 87.80 | 32.0 |
| SCoT | 89.55 | 41.0 |
| SCoT+MEM | 89.08 | 42.0 |
| SCoT+KG | 87.43 | 37.0 |
| SCoT+KG+MEM | 88.85 | 50.0 |
| MEDCOG-ORACLE | **98.98** | **67.0** |

We quantify inference efficiency via introducing two metrics: *Inference Density* and *Inference Incremental Efficiency (IIE)*. Experiments demonstrate that MEDCOG-META achieves superior performance and the highest IIE on 5 hard sets of medical benchmarks. MEDCOG-META achieves $6.2\times$ Inference Density as shown in Figure 1 (b), denoting the theoretical cost is $6.2\times$ of our actual cost, validating the mitigation of diminishing returns under scaling law via meta-cognitive regulation. More analyses reveal the meta-cognitive characteristics of LLMs and the necessity to exploit meta-recognition regulation. Our contributions are summarized as follows:

- We propose MEDCOG, the first comprehensive exploration of medical meta-cognition of LLMs to enable LLM-centric on-demand reasoning, where the assessments of complexity, familiarity, and knowledge Density dynamically activate procedural, episodic, and factual knowledge.

- We introduce *Inference Density* and *IIE* to quantify both inference effectiveness and efficiency. We empirically reveal the dual accuracy-cost benefits of meta-cognitive regulation and its mitigation to the inference scaling law. We also reveal the promising theoretical upper bound with the Oracle study.

- Experiments validate the effectiveness and efficiency of MEDCOG, providing insights including: meta-cognition characteristics across various LLMs and datasets, synergy of different knowledge types, fine-grained threshold activation, and thorough error analysis. These findings further highlight the critical role of LLM meta-cognitive regulation.

## 2. Pilot Analysis

To better motivate our study, we begin with a pilot study comparing different reasoning strategies in Table 1. We also explore the inference scaling law as shown in Figure 1, which is detailed in Section 3.4.

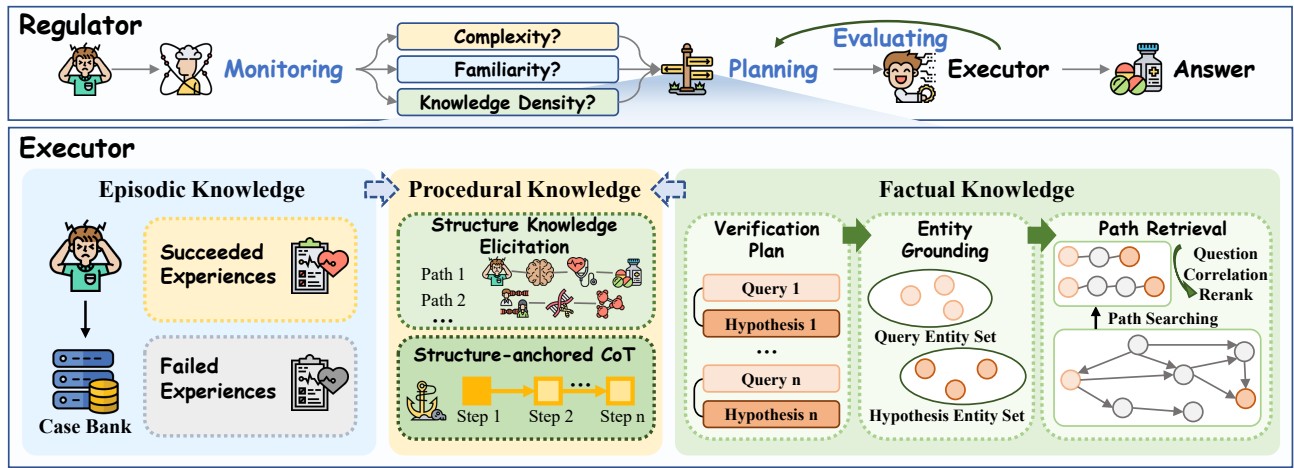

*Figure 2.* The MedCoG framework for medical reasoning, composed of (1) **Meta-Cognition Regulator** to route the reasoning strategies via Monitoring, Planning, and Evaluating. (2) **Knowledge Executor**, to provide Episodic (*Recalling Experiences*), Procedural (*Knowing How*), and Factual Knowledge (*Knowing What*). Based on its assessment of complexity, familiarity, and knowledge density, the regulator dynamically routes reasoning over different knowledge types.

**Experimental setting.** We design 5 different strategies to simulate different kinds of knowledge, detailed in Section 3.3: *(1)* ZERO-SHOT *reasoning*, where the LLM outputs the answer without thinking. *(2) Structural Chain-of-Thought* (SCOT), where the LLM first outputs the elicited structural paths, then outputs the step-by-step reasoning process. *(3) Memory-augmented SCoT* (SCOT+MEMORY), where the historical relevant experience, including question and reasoning trajectory, are provided. *(4) Knowledge Graph (KG) augmented SCoT* (SCOT+KG), where the retrieved knowledge graph paths are provided. *(5) Memory and KG augmented SCoT* (SCOT+KG+MEM), where both memory and KG are provided to the LLM to generate CoT reasoning.

We then introduce MEDCOG-ORACLE, which selects the optimal strategy for each sample from the five candidates. Specifically, for each case, MedCoG-Oracle represents whether any of the strategies in the strategy pool provides the correct answer, representing the empirical upper bound. We calculate the cost of MEDCOG-ORACLE in Figure 1 (a). To illustrate the costs of achieving oracle performance using different strategies, we plot a line between *min cost* and *max cost* as a reference.

**Observations.** (1) *Surprising Oracle Performance:* The ORACLE achieves 98.98 accuracy on the MEDQA-FULL set, surpassing current state-of-the-art (Vals AI, 2026) o1 (96.52). (2) *More is Less:* Simply augmenting the LLM with external memory or knowledge graph paths does not guarantee performance improvements; some configurations perform even worse than the ZERO-SHOT baseline. (3) *Low Cost of Oracle:* As shown in Figure 1 (a), even the max cost of Oracle is lower than AFlow with the highest accuracy, while the min cost only slightly surpasses CoT.

**Insights.** (1) *Clever Regulation over Blind Scaling:* The high oracle performance indicates that the reasoning ability of LLMs with incorporated three knowledge types shows high potential for medical reasoning. The bottleneck lies in case-wise strategy regulation, rather than blind scaling. (2) *Targeted Knowledge rather than All Knowledge:* The performance degradation of KG or memory suggests that for foundation models with considerable intrinsic knowledge, it is crucial to provide the targeted knowledge LLM requires, rather than all relevant knowledge. (3) *Win-win via regulation:* The ideal reasoning strategy is to approximate the Oracle, where LLM exactly knows which strategy is the best for itself, which can bring gains in both effectiveness and efficiency.

## 3. Method

### 3.1. Overview

Driven by the above insights, we propose the MEDCOG framework, a Medical Meta-Cognition Agent with Knowledge Graph. As shown in Figure 2, the meta-cognition regulator governs the reasoning pipeline via: (1) *Monitoring*: assessing the question in three dimensions, Complexity, Familiarity, and Knowledge Density. (2) *Planning*: Based on the assessments, the Regulator plans the execution of activated reasoning strategies, including procedural, episodic, and factual knowledge. For factual knowledge, the Regulator further plans the KG verification targets to elicit the demanded knowledge of LLMs. (3) *Evaluating*: After knowledge is gathered, the Regulator evaluates whether the Regulator is ready to output the answer or requires additional knowledge. MEDCOG governs over three types of knowledge: (1) *Procedural Knowledge*, the Structural

Chain-of-Thought (SCoT) that guides step-by-step reasoning through structural knowledge elicitation and anchoring. (2) *Episodic Knowledge*, which replays reasoning experiences from similar cases. (3) *Factual Knowledge*, retrieved paths from medical knowledge graphs that validate specific queries and hypotheses.

## 3.2. Meta-Cognition Regulator

**Monitoring:** Given a medical question $\mathcal{Q}$, the Monitoring module first inspects the contexts to derive a three-dimensional meta-cognitive state vector $\mathbf{s} = [s_c, s_f, s_k]$:

- Complexity $s_c$: Assesses the structural complexity of the query, indicating whether multi-hop reasoning or logical decomposition is required.
- Familiarity $s_f$: Assesses the level of typicality of the case. High familiarity indicates a standard, textbook-style case that is likely to align with historical reasoning patterns, while low familiarity indicates a rare case where past experiences may not apply.
- Knowledge Density $s_k$: Assesses the dependency on specific medical facts, such as entities or relationships. High knowledge density indicates the need for external facts or concept understanding to answer the question, while low density indicates sufficient knowledge.

**Planning:** Based on the state vector $\mathbf{s}$, the planning module acts as a policy network to select the optimal reasoning strategy $\mathcal{M} = \pi(\mathbf{s})$. We implement the policy network as a non-parametric gate with thresholds $\tau = [\tau_c, \tau_f, \tau_k]$ calibrated on the held-out validation set. We define binary indicator $I_j = \mathbb{1}(s_j > \tau_j)$ for each dimension. The selected strategy is defined in Equation (1):

$$\mathcal{M} = \pi(\mathbf{v}; \tau)$$
$$= \begin{cases} \text{ZERO-SHOT}, & \text{if } I_c + I_f + I_k = 0 \\ \text{SCoT} \oplus \sum_{j \in \{f,k\}} I_j \cdot \mathcal{M}_j, & \text{otherwise} \end{cases} \quad (1)$$

Specifically: (1) If all three scores fall below their thresholds, zero-shot reasoning is employed to directly answer the question. (2) Otherwise, structural CoT serves as the base reasoning method, optionally combined with MEMORY upon high familiarity and KG upon high knowledge density. The strategy space is thus $\mathcal{S} = \{\text{ZERO-SHOT, SCoT, SCoT+MEM, SCoT+KG, SCoT+KG+MEM}\}$. The meta-cognitive state $\mathcal{M}$ maps to a strategy $\mathbf{s} \rightarrow \mathcal{M}, \mathcal{M} \in \mathcal{S}$.

After mapping strategy $\mathcal{M}$, for KG-activated samples, the Planning module generates a KG verification plan that constrains the KG searching space based on the specific knowledge the LLM requires. In this way, the Regulator avoids noise from untargeted retrieval in an enormous searching space. In contrast, SCoT relies on the internal reasoning of LLMs based on the provided input and thus is self-contained. The Memory retrieval is based on task similarity and thus is pre-determined. Therefore, the Planning module routes to the memory retrieval function call and SCoT prompt without further planning for episodic and procedural knowledge.

**Evaluating:** To address the potential for retrieval failures within the Knowledge Graph (KG), we implement the Evaluating module after the Executor collects retrieved knowledge paths for verification. Since KG paths may occasionally be empty or provide insufficient evidence, the Evaluating module specifically scrutinizes the retrieved factual knowledge to determine if it is sufficient to support final answer delivery. If the retrieved KG path is insufficient, the Regulator could proceed to either (1) refine the plan for next-cycle retrieval or (2) activate SCoT reasoning to finalize the answer, integrating current evidence and its internal knowledge. The plan refinement is constrained in two cycles to prevent computational overhead.

## 3.3. Knowledge Executor

**Procedural Knowledge: *Knowing How*** For questions with at least one positive indicator out of three, we introduce Structural Chain-of-Thought (SCoT) Reasoning (Wei et al., 2022; Wu et al., 2025a) as the base reasoning strategy. Since Medical knowledge reasoning usually involves specialized concepts and their correlations to infer the final answer, we thus decouple Factual Knowledge from the reasoning process, formulated as $\text{SCoT} = (\mathcal{P}^e, \mathcal{C})$: the LLM is prompted to first elicit the structural facts as entity-relation paths $\mathcal{P}^e$ from its parameters or from optionally activated KG retrieval; then produces the reasoning procedural chain $\mathcal{C}$ to conclude the answer. This explicit decoupling enforces a structural-anchored CoT trajectory, bridging the gap between "Knowing What" and "Knowing How".

**Factual Knowledge: *Knowing What*** For questions with high knowledge density, where precise domain facts are required, we introduce a Knowledge Graph-based Verifier mechanism. Clinical questions typically contain extensive contextual details, such as various physiological indicators, yet only a specific subset is *critical* for reasoning. Direct retrieval from question $\mathcal{Q}$ may introduce noise from irrelevant contexts and expand the KG searching space, rather than delivering the knowledge the LLM actually requires.

*Knowledge verification plan:* Therefore, instead of using the raw question $\mathcal{Q}$, we employ a verification planning step to identify the specific knowledge required. We decompose the question $\mathcal{Q}$ into a set of verification pairs $\mathcal{V}(\mathcal{Q}) = \{(v_i, h_i)\}$, where $v_i$ denotes the atomic verification query and $h_i$ denotes the LLM's hypothesis. This allows us to focus on verifying the on-demand factual knowledge and narrow down the search scope.

*Medical knowledge graph:* We use a medical knowledge graph (Chandak et al., 2023) for factual knowledge retrieval.

A knowledge graph $\mathcal{G} = \{\mathcal{E}, \mathcal{R}, \mathcal{T}\}$ consists of an entity set $\mathcal{E}$, a relation set $\mathcal{R}$, and a triple set $\mathcal{T} \subset \mathcal{E} \times \mathcal{R} \times \mathcal{E}$ representing the relations between entities. The Medical KG provides structured factual anchors and evidence that ground the reasoning in specific facts to assist reasoning.

*Medical entity grounding:* For knowledge graph-based retrieval, we extract entities from the verification query $\mathcal{E}_v, \mathcal{E}_h = \text{Extractor}(v, h)$ through a three-step process: (1) A KG-grounding LLM (KG-LLM) extracts candidate entity phrases from $(v, h)$ as $\mathcal{E}_v, \mathcal{E}_h$. (2) Each candidate entity $e$ is then matched to its the most similar entity in KG $\mathcal{E}$ for each entity $e \in \mathcal{E}_x, x \in \{v, h\}$. The entity grounding procedure is denoted as Equation (2).

$$\hat{e} = \arg\max_{e_i^g \in \mathcal{E}} \text{Ranker}(e, \mathcal{E}; \theta)$$
$$= \arg\max_{e_i^g \in \mathcal{E}} \text{sim}(\text{enc}_\theta(e), \text{enc}_\theta(e_i^g)) \quad (2)$$

(3) The KG-LLM further refines the grounded entity set $\{\hat{e}\}$ in the verification-hypothesis pair based on context relevance, yielding $\hat{\mathcal{E}}_v, \hat{\mathcal{E}}_h = \text{KG-LLM}(\{\hat{e}_v\}, \{\hat{e}_j\})$.

*Path searching and ranking:* We retrieve the shortest path from the KG to link all verification pairs $(v_i, h_i) \in \mathcal{V}$, as Equation (3), where SP denotes shortest path searching.

$$\mathcal{P}^g(\mathcal{Q}) = \bigcup_{(v_i, h_i) \in \mathcal{V}(\mathcal{Q})} \{\text{SP}(e_v, e_h) \mid e_v \in \hat{\mathcal{E}}_{v_i}, e_h \in \hat{\mathcal{E}}_{h_i}\}, \quad (3)$$

To handle samples with numerous relevant paths, we rank all paths by their similarity to the original question $\mathcal{Q}$ as Equation $\tilde{\mathcal{P}}^g(\mathcal{Q}) = \text{TopK}_{p_j^g \in \mathcal{P}^g} \text{Ranker}(\mathcal{Q}, \mathcal{P}^g; \theta)$, where ranking with $\mathcal{Q}$ provides a global perspective from the original task context.

**Episodic Knowledge: *Recalling Experiences*** For familiar but complex questions, the memory from historical reasoning experiences can provide situational reasoning patterns for reference to solve new tasks. Thus, we construct an Episodic Knowledge Case Bank $\mathcal{B} = (q_i, (\mathcal{P}_i^e, \mathcal{C}_i), r_i)$ from the training set, where $q_i$ denotes the $i$-th question, $(\mathcal{P}_i^e, \mathcal{C}_i)$ denotes its reasoning experience in SCoT form, and $r_i \in \{0, 1\}$ denotes the reward to the reasoning process SCoT$_i$ if yielding the correct answer for question $q_i$. For memory retrieval, we retrieve similar cases from $\mathcal{B}$ by ranking each question $q_i$ in $\mathcal{B}$ based on its similarity to the current question $\mathcal{Q}$ as $\tilde{\mathcal{B}}(\mathcal{Q}) = \text{TopK}_{(q_i, \cdot) \in \mathcal{B}} \text{Ranker}(\mathcal{Q}, \mathcal{B}; \theta)$.

### 3.4. Inference Effective-Efficiency Analysis

**The Pareto Frontier Analysis:** As shown in Figure 1 (a), we plot the accuracy against inference cost (log scale) and establish the Pareto Frontier of existing approaches, where the curve represents the optimal performance achievable at any given cost. For "Pareto w/o ours", most existing methods increase computational cost but yield progressively

smaller marginal gains. In contrast, in "Pareto w/ ours", our MEDCOG-META significantly advances this frontier by regulating different reasoning strategies. Furthermore, the substantial gap between the current approaches and MEDCOG-ORACLE (upper bound) reveals the potential for improvement through meta-cognitive regulation.

**Inference Density:** LLM inference efficiency has been explored from multiple perspectives (Chen et al., 2026; Luo et al., 2026; Snell et al., 2024). Inspired by (Xiao et al., 2025), we introduce ***Inference Density*** to quantify inference efficiency. We first model the Scaling Curve of existing methods by fitting their accuracy $Acc$ and computational cost $C$ (USD per sample) to a logarithmic law, as shown in Equation (4), where $\alpha$ and $\beta$ are coefficients fitted from reference models.

$$Acc = f(C) = \alpha \cdot \ln(C) + \beta \quad (4)$$

We set the reference models as the methods from the baseline Pareto Frontier ("Pareto w/o ours") in Figure 1 (a) in implementation. As shown in Figure 1 (b), the fitted curve achieves $R^2 = 0.996$, indicating that existing methods on 5 hard sets align largely with this scaling curve.

We define the Inference Density $\rho$ for a model $\mathcal{M}$ as the ratio of effective cost to actual cost, as shown in Equation (5). The effective cost $\hat{C}(Acc_\mathcal{M})$ represents the cost that the reference models would require to achieve the same accuracy as model $\mathcal{M}$, i.e., the projected cost on the scaling curve achieving accuracy $Acc_\mathcal{M}$.

$$\rho(\mathcal{M}) = \frac{\hat{C}(Acc_\mathcal{M})}{C_\mathcal{M}} = \frac{f^{-1}(Acc_\mathcal{M})}{C_\mathcal{M}} \quad (5)$$

As illustrated in Figure 1 (b), MEDCOG-META achieves an Inference Density of $6.2\times$, meaning it delivers performance equivalent to a reference model consuming 6.2 times of MEDCOG-META's cost. This confirms that meta-cognitive regulation effectively compresses the reasoning cost, mitigating the diminishing returns of the inference scaling law.

**Inference Incremental Efficiency:** While Inference Density provides a global perspective relative to the fitted scaling curve composed of existing methods, it inherently relies on the distribution of reference models. To offer a more direct assessment of inference efficiency independent of curve fitting, we introduce a second metric: Inference Incremental Efficiency (IIE).

In the health economy field, the Incremental Cost-Effectiveness Ratio (ICER) (Weinstein & Stason, 1977) is a standard metric for evaluating efficiency of interventions, representing the incremental cost required to achieve one additional unit of health effect. ICER emphasizes the "cost of improvement" ($\downarrow$) rather than the "efficiency of cost" ($\uparrow$). Inspired by ICER, we define Inference Incremental Efficiency (IIE) as the Effectiveness-Cost Ratio (reciprocal of

ICER), representing the marginal accuracy gain per additional unit of inference cost. Given that Chain-of-Thought (CoT) serves as the primary reasoning paradigm in current research, we establish CoT as the baseline reference ($\mathcal{M}_0$). The IIE is formally defined as Equation (6), where $Acc_{\mathcal{M}}$ represents the accuracy of method $\mathcal{M}$ on medical reasoning benchmarks, and $C_{\mathcal{M}}$ is the inference cost measured in USD per sample.

$$\text{IIE}_{\mathcal{M}} = \frac{Acc_{\mathcal{M}} - Acc_{\mathcal{M}_0}}{C_{\mathcal{M}} - C_{\mathcal{M}_0}}, \tag{6}$$

In this formulation, each reasoning method is viewed as an *intervention* applied to the base LLM+CoT process, highlighting the *incremental capability in addition to the intrinsic capability of LLMs*. A higher IIE thus indicates better marginal efficiency in translating additional computational cost into reasoning performance gains. It is also worth noting that IIE is the gradient of the line from CoT to $\mathcal{M}$ in Figure 1 (b). We report IIE in Table 2.

We adopt CoT as the baseline reference for the following reasons: (1) CoT serves as the standard reasoning paradigm adopted by all compared methods, ensuring a fair comparison of incremental gains; (2) using Zero-Shot as the baseline would conflate the gains from CoT reasoning itself with the gains from the method, obscuring the true incremental efficiency of each approach; (3) considering the minimum necessary cost for each case, including the question and contexts, CoT represents the minimum necessary and reasonable expenditure for reasoning, and subtracting its cost isolates the *additional* cost incurred by each method.

## 4. Experiments

### 4.1. Datasets

We conduct experiments on complex medical reasoning benchmarks, using hard subsets extracted by MedAgentsBench (Tang et al., 2025). These hard subsets comprise questions on which **more than 50% of LLMs fail to answer correctly**. We evaluate our method on the hard subsets of widely adopted benchmarks: MedQA (Jin et al., 2021), MedMCQA (Pal et al., 2022), MMLU (Hendrycks et al., 2020), MMLU-Pro (Wang et al., 2024a), PubMedQA (Jin et al., 2019). Additionally, we evaluate on the MedQA full set, since it is the most representative benchmark, encompassing questions of various levels of complexity, and thus providing a comprehensive assessment. The MMLU-Hard contains 73 samples, while the other hard subsets each contain 100 samples. The MedQA-Full set contains 1,273 samples. All datasets are formatted as multiple-choice questions with candidate options provided.

### 4.2. Baselines

We compare our method with three groups of baselines: *(1) General Reasoning Baselines*: Zero-Shot, Few-Shot, CoT (Wei et al., 2022), CoT-SC (Wang et al., 2022). *(2) General Reasoning Agent*: MultiPersona (Wang et al., 2024b), Self-Refine (Madaan et al., 2023), AFlow (Zhang et al., 2025a). *(3) Medical Agent*: MedPrompt (Nori et al., 2023b), MedAgents (Tang et al., 2024), MDAgents (Kim et al., 2024). Accuracy (%) and IIE (Equation (6)) are reported for quantitative comparison. The accuracy results and cost per sample ($) are as reported in MedAgentsBench (Tang et al., 2025), where all results are based on GPT-4o backbone.

### 4.3. Implementation Details

We employ GPT-4o (Hurst et al., 2024) (2024-08-06) as the backbone LLM for Meta-Cognition Regulator and SCoT reasoning, and GPT-4o-mini (Hurst et al., 2024) (2024-07-18) for KG grounding, both accessed via the OpenAI API. We set the temperature to 0 to ensure reproducibility. For the ranker model $\theta$ used in entity grounding, path ranking of Factual Knowledge, and memory retrieval of Episodic Knowledge, we use the bge-base-en-v1.5 model. The meta-cognition regulator thresholds $\tau = [\tau_c, \tau_f, \tau_k]$ are calibrated using 50 random samples from the training set (non-overlapped with evaluation data). For factual knowledge, we employ PrimeKG (Chandak et al., 2023), consistent with MedReason (Wu et al., 2025a), which covers over 4 million relationships across major medical and biological domains. We set $K = 5$ for path ranking and $K = 5$ for case bank retrieval. For case bank construction, we utilize MedReason (Wu et al., 2025a) with structural fact elicitation and quality filtering. Specifically, we filter the MedReason CoT data with only structural paths, ensuring decoupled facts and reasoning, and forming unified SCoT data. For MedQA and MedMCQA, we construct the case bank $\mathcal{B}$ with 1,402 and 1,305 SCoT cases, respectively. For MMLU, MMLU-Pro, and PubMedQA, we merge the case banks from MedQA and MedMCQA, yielding 2,707 cases in total. This setup allows us to evaluate the transferability of episodic knowledge. GPU usage is less than 2GB for accelerating memory and KG retrieval.

### 4.4. Main Results

The comparison results on hard subsets of medical benchmarks are shown in Table 2.

**Comparison to Baselines:** As shown in Table 2, MEDCOG-META achieves the best average accuracy and outperforms the second-best baseline AFlow by 8.7%, demonstrating the overall effectiveness on complex medical reasoning. Notably, the general reasoning baselines surprisingly outperform agentic methods on some datasets, suggest-

*Table 2.* Performance of existing approaches on Hard Subsets from MedAgentsBench ([Tang et al., 2025](#)), questions in which $> 50\%$ LLMs cannot infer correctly. IIE* denotes we scale the $C$ in Eq. (6) from per sample to per 1k samples for readability. MEDCoG-ALL denotes "SCoT+KG+Memory" with three types of knowledge.

| METHOD | MEDQA | MEDMCQA | MMLU | MMLU-PRO | PUBMEDQA | AVG. ACC. | IIE* (↑) |
|---|---|---|---|---|---|---|---|
| ZERO-SHOT | 32.0 | 25.0 | 24.7 | 21.0 | 9.0 | 22.3 | - |
| FEW-SHOT | 28.0 | 29.0 | 27.4 | 9.0 | **20.0** | 22.7 | - |
| CoT | 39.0 | 30.0 | 26.0 | 35.0 | 10.0 | 28.0 | REF |
| CoT-SC | 37.0 | 35.0 | 30.1 | 43.0 | 6.0 | 30.2 | 0.111 |
| SELF-REFINE | 41.0 | 34.0 | 34.2 | 34.0 | 13.0 | 31.2 | 0.345 |
| MULTIPERSONA | 45.0 | 25.0 | 37.0 | 42.0 | 15.0 | 32.8 | 0.162 |
| AFLOW | 48.0 | 31.0 | **38.4** | 37.0 | 18.0 | 34.5 | 0.141 |
| MEDPROMPT | 34.0 | 26.0 | 26.0 | 22.0 | 11.0 | 23.8 | -0.208 |
| MEDAGENTS | 43.0 | 30.0 | 28.8 | 8.0 | 15.0 | 25.0 | -0.035 |
| MDAGENTS | 36.0 | 22.0 | 24.7 | 8.0 | 11.0 | 20.3 | -0.165 |
| MEDCoG | W/ IN-DISTRIBUTION MEMORY | | W/ OUT-OF-DISTRIBUTION MEMORY | | | | |
| MEDCoG-META | **52.0** | **36.0** | 35.6 | **44.0** | **20.0** | **37.5** | **0.438** |
| MEDCoG-ALL | 50.0 | 32.0 | 28.8 | 36.0 | 19.0 | 33.2 | 0.181 |

*Table 3.* Per-dimension Precision/Recall/F1 of the Meta-Cognitive Monitor across backbones on MedQA-H.

| Monitor | Complexity | Familiarity | Knowledge |
|---|---|---|---|
| GPT-4o | 0.95 / 0.63 / 0.75 | 0.83 / 1.00 / 0.90 | 0.59 / 0.95 / 0.80 |
| GPT-4o-mini | 1.00 / 0.78 / 0.88 | 0.83 / 0.83 / 0.83 | 0.62 / 0.95 / 0.75 |
| Gemini-2.0-Flash | 0.97 / 1.00 / 0.98 | 0.88 / 0.83 / 0.86 | 0.69 / 0.95 / 0.80 |
| o3-mini | 0.97 / 0.88 / 0.92 | 0.65 / 1.00 / 0.79 | 0.68 / 0.72 / 0.70 |
| Qwen3-8B | 0.97 / 0.94 / 0.95 | 0.47 / 0.64 / 0.54 | 0.75 / 0.21 / 0.33 |
| Qwen3-32B | 0.97 / 0.91 / 0.93 | 0.76 / 1.00 / 0.87 | 0.67 / 1.00 / 0.80 |
| Qwen3-Max | 0.97 / 0.97 / 0.97 | 0.75 / 1.00 / 0.86 | 0.68 / 0.94 / 0.79 |

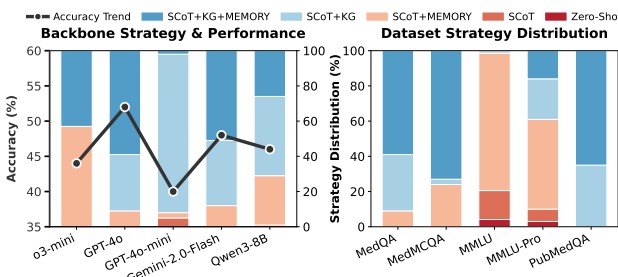

*Figure 3.* Strategy distribution (a) with different Monitoring backbones, and (b) on different datasets.

ing that costly reasoning strategies could introduce negative transfer that undermines the intrinsic knowledge of LLMs. Through our meta-cognition regulation, MEDCoG-META outperforms all general and agentic methods, demonstrating the effectiveness of adaptive, on-demand strategy selection based on the LLM's thorough assessments. Moreover, MEDCoG-META consistently outperforms MEDCoG-ALL with all three types of knowledge, demonstrating the necessity and effectiveness of meta-cognition regulation.

**Effect of Case Bank Domain:** On datasets where MED-CoG w/ OOD memory, MMLU, MMLU-Pro, and Pub-MedQA have different medical reasoning patterns with MedQA and MedMCQA in the Case Bank. Remarkably, MEDCoG-META achieves the best or second-best performance on these OOD datasets, demonstrating robust transferability of various medical reasoning patterns. Moreover, incorporating OOD memory may introduce negative impacts (detailed in Appendix K), which Meta-Cognition Monitoring effectively filters out with the familiarity score threshold. We provide a direct comparison of the In-Distribution versus OOD case banks on PubMedQA in Appendix J, as its Yes/No/Maybe reasoning format differs from other multi-choice datasets.

**IIE Analysis:** MEDCoG-META achieves the highest IIE

score, indicating that it yields the most significant accuracy gains per unit of computational cost among all methods. In contrast, the baselines largely follow the inference scaling law, requiring higher costs for greater accuracy gains, such as AFlow. Moreover, medical agentic methods fail to outperform CoT and thus obtain negative IIE values, where additional costs fail to produce higher accuracy, probably due to poor adaptivity of the multi-agent architecture to hard cases. Furthermore, the substantial IIE advantage of MEDCoG-META over MEDCoG-ALL demonstrates that meta-cognition regulation successfully maximizes the inference efficiency through on-demand strategy selection.

### 4.5. Meta-Cognition Analysis

We explore the meta-cognition strategy in several aspects: (1) Quantitative quality of meta-cognitive monitor across backbones (Table 3), (2) Strategy distributions across backbones (Figure 3 (a) and Appendix D), (3) Strategy distributions across datasets (Figure 3 (b) and Appendix D), (4) Meta-cognition score threshold study (Figure 4 and Appendix E), (5) Meta-Cognition Score distribution visualization on MedQA (Appendix C).

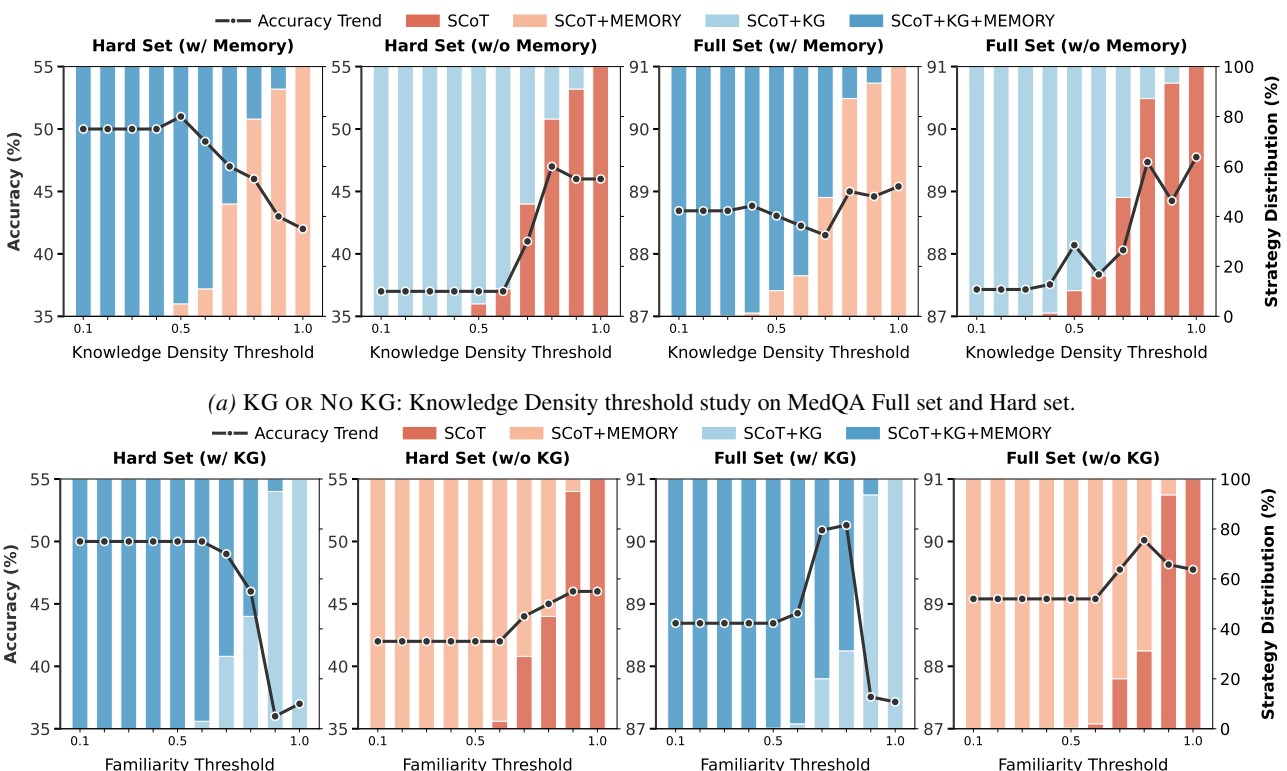

*(a)* KG OR NO KG: Knowledge Density threshold study on MedQA Full set and Hard set.

*(b)* MEMORY OR NO MEMORY: Familiarity threshold study on MedQA Full set and Hard set.

*Figure 4.* Meta-Cognition score threshold study of Knowledge Density and Familiarity on MedQA. Complexity study is in Figure 6.

**Meta-Cognitive Monitoring Accuracy.** To quantify the calibration quality of the meta-cognitive monitor, we report per-dimension Precision/Recall/F1 against oracle strategy labels in Table 3. All backbones demonstrate above-chance alignment, with Complexity and Familiarity F1 generally exceeding Knowledge Density. GPT-4o, GPT-4o-mini, and Gemini-2.0-Flash achieve high Knowledge Density recall but lower precision, indicating a tendency to over-activate KG retrieval. o3-mini attains perfect Familiarity recall but lower precision, confirming its overestimation of case typicality. Qwen3-8B collapses on Knowledge Density (F1=0.33), yet scaling up to Qwen3-32B and Qwen3-Max substantially recovers both Familiarity and Knowledge Density F1, demonstrating that larger models exhibit more reliable meta-cognitive monitoring.

**Strategy Distributions across Backbones.** We investigate the impact of different LLM backbones on the three-dimensional monitoring scores in Figure 3 (a). Results indicate that GPT-4o and Gemini-2.0-Flash share similar strategy distributions, with balanced activation across the three strategies. In contrast, advanced LLMs and smaller LLMs often exhibit problematic tendencies: overconfidence (o3-mini and Qwen3-8B underestimates the ratio of factual knowledge retrieval) and unfamiliarity (GPT-4o-mini fails to activate memory in most cases). These findings un-

derscore the necessity of investigating both the knowledge boundaries and the self-assessments of their cognitive state for LLMs.

**Domain-Adaptive Knowledge Demands.** Strategy distributions also vary significantly across datasets in Figure 3 (b). MEDCOG shows relatively high KG ratios for MedQA, MedMCQA, and PubMedQA, indicating complex, noisy contexts of these datasets requiring precise factual knowledge. Conversely, for generalized medical evaluation benchmark MMLU and MMLU-Pro, the system leans towards Episodic Memory, favoring case-based reasoning from episodic knowledge over fact retrieval. This domain-adaptive regulation confirms that MEDCOG can adjust its strategy for specific reasoning demands.

**Impact of Regulation Thresholds.** We conduct threshold studies to intuitively show at which levels LLMs benefit from the three types of knowledge sources, based on fine-grained regulation thresholds ($\tau_k, \tau_f, \tau_c$) as shown in Figures 4 and 6. The performance trends of three thresholds further reveal that indiscriminate addition or elimination of any knowledge type degrades the performance, due to noise, over-reasoning, or insufficient knowledge. Crucially, a synergy effect is observed: combining Memory would boost the performance of KG retrieval, especially on hard sam-

*Table 4.* Error Type Statistics. Definitions are in Appendix I.2.

|  | $\mathcal{S}$ | MedCoG-Meta |
|---|---|---|
| Feeling of Knowing | 13 | 4 |
| Over Reasoning | 33 | 14 |
| Synergy Missed | 29 | 4 |
| Memory Noise | 20 | 3 |
| KG Noise | 23 | 10 |
| Information Conflict | 5 | 2 |
| Unsolvable | 33 | 33 |

ples. This suggests that episodic memory facilitates comprehension of abstract knowledge graph paths, validating the design of our co-adaptive execution mechanism. Moreover, only high-familiarity samples benefit from Memory, while factual knowledge is more critical for hard samples. More detailed analysis is provided in Appendix. E.

### 4.6. Error Analysis

We conduct an error analysis on the results of strategy pool $\mathcal{S}$ and MEDCOG-META on MedQA-Hard, as shown in Table 4. The definitions for each error type and case examples are detailed in Appendix I.

Quantitatively, the Regulator achieves a global routing accuracy of 52/67 = 77.6% on MedQA-H, with the remaining 22.4% attributed to suboptimal strategy allocation. As shown in Table 4, MEDCOG-META effectively mitigates the Feeling of Knowing and Synergy Missed by activating on-demand knowledge accordingly, while reducing Memory/KG Noise and Information Conflict by controlling the involvement of unnecessary knowledge. Furthermore, (1) Low Feeling-of-Knowing and high Over-Reasoning cases indicate that SCoT or zero-shot reasoning would be enough for 33 samples, while only 13 samples would benefit from additional knowledge. It underscores the necessity of meta-cognitive calibration to increase confidence in its proficient knowledge. (2) KG noise is relatively high since KG paths tend to be noisy and abstract for LLMs to understand. Improving structural knowledge comprehension is critical for KG-augmented LLM, and its synergy with episodic memory presents a promising direction. (3) More effective strategies could further raise the upper bound of strategy pools by reducing the 33 unsolvable cases.

## 5. Conclusion

In this paper, we propose MEDCOG, a meta-cognition medical agent with knowledge graph, to investigate how the self-assessments of LLMs can regulate the reasoning process. We empirically explore the inference scaling law and introduce two metrics to quantify inference efficiency,

Inference Density and Inference Incremental Efficiency. Experiments demonstrate the inference effectiveness and efficiency of MEDCOG. Our main findings are as follows: (1) Meta-cognition regulation maximizes the inference density, improving both accuracy and efficiency. (2) Different LLMs exhibit distinct meta-cognitive characteristics; per-backbone calibration is essential to bring out their best performance. (3) The memory mechanism shows a synergy effect with KG, implying that memory can improve the understanding of structural yet noisy factual knowledge of LLMs. Our study reveals the gap between existing methods and the upper bound of meta-cognitive regulation, motivating future research for LLM meta-cognition regulation.

## Acknowledgements

This research is supported by the Singapore Ministry of Health through the National Medical Research Council (NMRC) Office, MOH Holdings Pte Ltd under the NMRC SIMFONI Funding Initiative (MOH-002026).

## Impact Statement

The development of MEDCOG offers significant broader impacts across the machine learning and healthcare landscapes.

**LLM Inference:** By introducing meta-cognitive regulation, our work provides critical insights into inference scaling, demonstrating that activating a model's intrinsic self-assessments can effectively maximize the inference density and mitigate the diminishing returns under scaling laws.

**Intuitive Demonstrations of LLM Meta-Cognition:** Our empirical findings further the understanding of LLM self-assessments, showing the necessity of LLM-centric on-demand reasoning, which reduces negative impacts and aligns external knowledge to LLM's specific needs.

**Economic Medical AI Application:** Our inference efficiency and interpretable meta-cognition processes could benefit practical medical AI applications. Inference efficiency improves the economic feasibility of the Medical AI Agent. Moreover, by assessing the LLM's self-assessments and reasoning process, the medical specialists could critically evaluate the credibility of the LLM's output.

**Ethical Considerations:** MEDCOG is designed as an assistant tool for augmenting rather than replacing the professional medical reasoning process. Furthermore, the performance is inherently correlated to the quality of the Case Bank and Knowledge Graph, which may lead to potential biases from data sources. The hallucinations of LLMs are also unavoidable due to black-box parameters. Thus, the user should remain vigilant for such application scenarios.

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

## Appendix Outline

## A. Related Work

### A.1. Knowledge-augmented Medical Reasoning

Recent advancements have significantly enhanced medical reasoning by injecting different forms of knowledge into Large Language Models (LLMs). We categorize related studies based on the incorporated knowledge of their methods.

*Procedural Knowledge* studies focus on improving the logical reasoning process itself. For instance, HuatuoGPT-o1 (Chen et al., 2024a) employs Chain-of-Thought (CoT) tuning to elicit complex medical reasoning capabilities. To provide facts to guide the CoT tuning, MedReason (Wu et al., 2025a) and Senator (Wei et al., 2025) leverage Knowledge Graphs (KGs) to construct structural CoT data for supervised fine-tuning and distillation, effectively improving the logical multi-hop medical reasoning ability of 7-8B-sized LLMs. *Factual Knowledge* studies address the hallucination of medical facts through retrieval mechanisms (Xu et al., 2025b). Approaches like MedGraphRAG (Wu et al., 2025b) and RAR$^2$ (Xu et al., 2025a) utilize retrieval techniques to ground LLM generation in evidence-based medical corpus, including knowledge graphs, documents, papers, etc., ensuring factual precision. *Episodic Knowledge* studies utilize historical experiences to guide current reasoning (Nori et al., 2023a). MedPrompt (Nori et al., 2023b) introduced kNN-based few-shot exemplar retrieval combined with ensemble voting to steer the model. Furthermore, MDTeamGPT (Chen et al., 2025) constructs both CoT and Correct Answer knowledge bases to support reasoning. DoctorRAG (Lu et al., 2025) retrieves from a clinical knowledge base and a patient base to provide episodic context. However, these methods often employ static reasoning pipelines without assessing the specific knowledge requirements of the query.

### A.2. Agentic Medical Reasoning

Recently, agentic frameworks simulate clinical workflows through multi-agent collaboration, iterative refinement, tool utilization, and RAG. MedPrompt (Nori et al., 2023b) laid the groundwork by combining prompting strategies into a unified pipeline. Subsequently, multi-agent frameworks have emerged to tackle complex queries. MedAgents (Tang et al., 2024) gathers expert agents to engage in iterative discussion until a decision is reached, while ReConcile (Chen et al., 2024b) introduces a debate mechanism among multi-family LLMs to foster a convincing reasoning process. To improve adaptability, MDAgents (Kim et al., 2024) dynamically recruit a discussion team based on a preliminary complexity check, and KAMAC (Wu et al., 2025c) incorporates knowledge gap detection within the multi-agent interaction. Other works focus on integrating

*Table 5.* Comparison of MEDCOG with existing reasoning frameworks across meta-cognitive dimensions. MEDCOG uniquely integrates comprehensive knowledge types with a training-free, instance-wise regulation.

| Method | Meta-Cognition Regulation Type | Knowledge Type | Training-Free | Instance-wise Strategy |
|---|---|---|---|---|
| ReAct (Yao et al., 2022) | Planning | Procedural | ✔ | ✗ |
| Self-Refine (Madaan et al., 2023) | Evaluating | Procedural | ✔ | ✗ |
| Reflexion (Shinn et al., 2023) | Evaluating | Procedural | ✔ | ✗ |
| AFlow (Zhang et al., 2025a) | Plan. & Eval. | Procedural | ✗ (Workflow Optim.) | ✗ (Dataset-level) |
| MetaRAG (Zhou et al., 2024) | Moni. & Plan. & Eval. | Procedural, Factual | ✔ | ✔ |
| RGAR (Liang et al., 2025) | Evaluating | Factual | ✔ | ✗ |
| MCTR (Li et al., 2025) | Planning | Procedural | ✗ (SFT+TTRL) | ✔ |
| KnowSelf (Qiao et al., 2025) | Monitoring | Procedural, Factual | ✗ (SFT) | ✔ |
| **MedCoG (Ours)** | **Moni. & Plan. & Eval.** | **Proc. + Fact. + Epis.** | ✔ | ✔ |

clinical tools and environments. ColaCare (Wang et al., 2025) incorporates Electronic Health Records (EHR) into the reasoning loop, and DrAgent (Liu et al., 2025) automates the workflow through specialized tool usage. AMG (Rezaei et al., 2025) enables agentic knowledge graph RAG and achieves autonomous medical KG construction and continuous updating. Moreover, interactive medical RAG frameworks like AgentClinic (Schmidgall et al., 2024), AgentHospital (Li et al., 2024a), and Mediq (Li et al., 2024b) simulate the entire clinical decision-making process in virtual hospital environments. While demonstrating high potential in real-world clinical tasks, these agentic methods often incur prohibitive inference costs due to indiscriminate test-time scaling, lacking an on-demand meta-cognitive mechanism that leverages the LLM's intrinsic self-awareness to synergize heterogeneous knowledge types.

### A.3. Meta-Cognition for LLM Agent

The integration of meta-cognition of LLM-based agents, i.e., the ability to monitor and regulate their own reasoning process, has emerged as a pivotal direction for enhancing LLM agents. The comparison between MedCoG and meta-cognitive agentic studies is listed in Table 5. In earlier studies, ReAct (Yao et al., 2022) regulates between reasoning and acting via static pipelines, lacking explicit self-regulation mechanisms. To improve adaptivity, recent works have adopted the Reactive Paradigm. Specifically, Reflexion (Shinn et al., 2023) and Self-Refine (Madaan et al., 2023) introduce iterative self-correction loops, where the model generates an initial response and then *reactively* refines it based on post-hoc critique. AFlow (Zhang et al., 2025a) automates agentic workflow optimization through dataset-level search rather than instance-level dynamic regulation. Such iterative reactive refinement workflows have also been adapted to retrieval. MetaRAG (Zhou et al., 2024) employs a "monitor-evaluate-correction" loop, where the model critiques its output or triggers re-retrieval *after* generation. Similarly, RGAR (Liang et al., 2025) utilizes recursive generation to iteratively refine retrieval relevance. However, these reactive methods inevitably incur sunk costs by executing suboptimal initial attempts before correction. Another line of research focuses on Internalization via Training. KnowSelf (Qiao et al., 2025) utilizes supervised fine-tuning (SFT) to align models with internal uncertainty states for switching between "fast" and "slow" modes, while MCTR (Li et al., 2025) enables agents to learn reasoning patterns from test-time trajectories. Despite their adaptiveness, these approaches rely on expensive training or implicit black-box parameters, limiting interpretability. In contrast to these reactive or training-heavy approaches, valid frameworks for pre-emptive and interpretable regulation remain underexplored.

## B. Datasets and Reasoning Domains

As shown in Table 6, each dataset emphasizes different reasoning aspects. It explains the distribution discrepancy in Figure 3: Specifically, MMLU and MMLU-Pro depend less on factual knowledge and more on the model's internalized parametric knowledge and experience imitation. This is attributed to their primary objective: assessing general domain understanding and robust fact recall inherent in LLM parameters. Conversely, datasets such as MedQA, MedMCQA, and PubMedQA prioritize the application of specialized professional knowledge to complex medical scenarios, such as clinical decision, knowledge reasoning, and context understanding, thereby necessitating external factual retrieval to formulate accurate clinical decisions.

## C. Score Distribution of Meta-Cognition Regulator

As shown in Figure 5, we visualize the distribution of meta-cognitive scores $s_c, s_f, s_k$ to understand the intrinsic decision boundaries of different backbones.

*Table 6.* Comparison of Medical Reasoning Benchmarks utilized in this study and their reasoning domains.

| Dataset | Source | Question Format | Options | Primary Reasoning Task |
|---|---|---|---|---|
| **MedQA** | USMLE (USA) | Clinical Contexts | 4 | Clinical decision. |
| **MedMCQA** | AIIMS/NEET (India) | Medical Knowledge | 4 | Medical knowledge reasoning and understanding. |
| **MMLU** Medical Subset | Exams & Textbooks | Medical Knowledge | 4 | Fact recall and basic clinical knowledge application. |
| **MMLU-Pro** Medical Subset | Scientific QA websites and exams | Clinical Contexts & Medical Knowledge | 10 | Fact recall and mult-hop reasoning, with multiple distractor options. |
| **PubMedQA** | PubMed Abstracts | Medical Paragraphs | 3 (Y/N/M) | Medical context understanding and reasoning. |

- **Complexity**: While most LLMs exhibit top-heavy distributions with median scores above 0.6, GPT-4o and Qwen3-8B distinctly differentiate two sets, assigning higher median complexity scores to the Hard set than the Full set. It indicates that: (1) Hard set shows a higher difficulty level, (2) GPT-4o and Qwen-3B show complexity sensitivity to capture such a distribution shift.

- **Familiarity**: GPT-4o-mini shows bottom-heavy distribution, implying it perceives most hard queries as novel challenges, thus less triggers the MEMORY strategy for imitation of historical patterns. On the contrary, o3-mini shows high familiarity with most queries, suggesting its ability to effectively assimilate episodic experiences and solve problems via analogical transfer rather than redundant computation.

- **Knowledge Density:** GPT-4o, Gemini-2.0-Flash, and GPT-4o-mini have similar distributions, while GPT-4o recognizes top knowledge density samples with $s_k > 0.8$. Qwen3-8B shows an incompact distribution, reflecting its unstable calibration. o3-mini exhibits distributions with overall lower knowledge density evaluation, which reflects its confidence in factual knowledge and may underestimate the necessity for fact retrieval when cooperating with less-advanced LLMs as executors.

## D. Detailed Analysis of Strategy Distribution

As shown in Figure 3, we investigate the strategy distributions with different LLM backbones for 3-dimensional score monitoring and on different datasets.

**Monitoring Backbones:** We replace the LLM in only the Monitoring process with other LLMs and keep the rest unchanged. As shown in Figure 3 (a), GPT-4o and Gemini-2.0-Flash achieve the best and the second-best performance, which also share similar strategy distributions with a high ratio of SCOT+KG+MEMORY strategy.

*For the more advanced model*, o3-mini is basically familiar with all reasoning patterns and thus triggers MEMORY in all cases. However, over-confidence on knowledge density cognition led to a low ratio of KG strategy and thus undermines the whole performance.

*For the smaller model*, the KG rate of GPT-4o-mini is similar to GPT-4o and Gemini-2.0-Flash. The key bottleneck for GPT-4o-mini is that it fails to trigger the MEMORY in most questions, attributed to low familiarity scores, which means the requirement of learning rather than imitation. However, most raw knowledge paths could be noisy and hard to understand without experience replay, which led to the low performance. Qwen3-8B underestimates of factual knowledge and activates MEMORY and KG in isolation for most cases, lacking synergy meta-cognition. Qwen series with larger sizes, Qwen3-32B and Qwen3-Max shows better meta-cognitive recall and F1 in Table 3, highlighting the meta-cognitive ability scales with model capacity and confirming the necessity of per-backbone calibration.

**Dataset Strategy Distributions:** For different datasets, the need for different knowledge varies based on reasoning domains. As shown in Figure 3 (b), MEDCOG-META is able to regulate knowledge execution strategies dynamically. The ratio of KG strategy is relatively high for MedQA, MedMCQA, and PubMedQA, which indicates more factual knowledge is

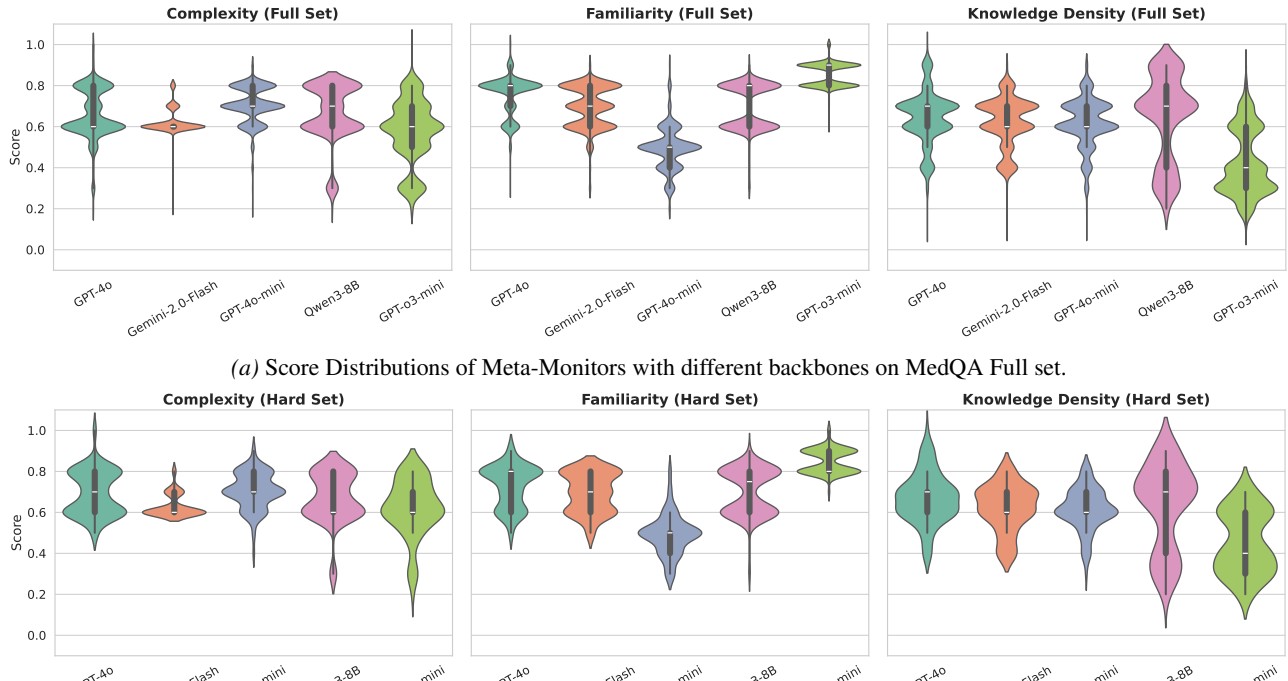

*(a)* Score Distributions of Meta-Monitors with different backbones on MedQA Full set.

*(b)* Score Distributions of Meta-Monitor with different backbones on MedQA Hard set.

*Figure 5.* Score Distributions of Meta-Monitor with different backbones on MedQA Full set and Hard set.

required for the clinical tasks involving complex contexts; while lower for MMLU and MMLU-Pro, which indicates less factual knowledge and more episodic experience imitation for the generalized medical knowledge assessment benchmarks. Moreover, MMLU-Pro activates factual knowledge more than MMLU, attributed to its challenging interrupting options.

## E. Threshold Study of Meta-Cognition Regulator

We explore the threshold of Complexity, Familiarity, and Knowledge Density in Figure 6, Figure 4b, and Figure 4a to bring more insights.

### E.1. KG OR NOT: Knowledge Density Threshold Study

**Retrieval Only Necessary:** The accuracy trends reveal that factual retrieval is beneficial only within specific confidence intervals, whereas indiscriminate retrieval would introduce noise and degrade reasoning performance. For the Full set, the model generally exhibits a lower demand for external facts and reaches similarly best performance around $\tau_k = 0.8$ and $\tau_k = 1.0$, confirming that extensive factual support is unnecessary for simpler queries. For the Hard set, the best performances with and without Memory are also achieved through a proportion of retrieval, rather than all retrieval or all non-retrieval.

**Synergy with Episodic Knowledge:** Notably, the strategies with Memory often brings stable and better performance for KG. When without Memory, more KG retrieval brings more negative impacts on both Hard and Full set, probably attributes to the structural path is not understood by the LLM. With Memory, the performance is generally improved. Especially on the Hard set where the questions naturally require more factual knowledge, adding Memory leads to the best performance. It highlights a synergy effect: episodic memory facilitates the comprehension of abstract knowledge graph paths, thereby amplifying the benefits of factual retrieval compared to using KG in isolation.

### E.2. MEMORY OR NOT: Familiarity Density Threshold Study

**Recall Experience when High Familiarity:** On a full set, memory retrieval yields positive gains only at high familiarity scores ($> 0.8$), suggesting that experience imitation is only effective when the LLM thinks the reasoning pattern in the certain question is typical and familiar. It is an analogy to real-world experience, i.e., recalling similar experience would be beneficial for a complex but "Deja vu" sample.

**Negative Impacts for Hard Cases:** On the hard set, the "Deja vu" questions are fewer since the hard questions usually involve rare knowledge or rare patterns, thus involving historical experiences would bring noise and downgrade the performance. In such case, retrieving factual knowledge or activate long thinking would be more effective. It further demonstrates the necessity of on-demand meta-cognition regulation.

### E.3. CoT OR NOT: Complexity Threshold Study

**Necessity for Structural CoT:** The general trend demonstrates that SCoT significantly outperforms DA, particularly on the Hard set, where the accuracy curve remains stable only when SCoT is dominant (orange regions), confirming that complex medical queries require rigorous step-by-step derivation.

**Over-reasoning Impacts:** A notable exception emerges on the Full Set, where the hybrid strategy at $\tau_c = 0.6$ yields higher accuracy than the pure SCoT baseline ($\tau_c = 0$). This indicates that for queries identified as low-complexity, the LLM's intuitive "System 1" response (DA) is superior. By bypassing SCoT for these simple cases, the meta-cognitive regulator effectively mitigates "over-reasoning", where forced logical decomposition of trivial questions can inadvertently introduce hallucinated complications.

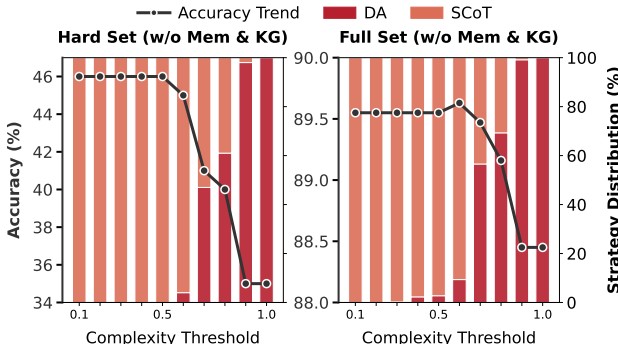

*Figure 6.* COT OR NO COT: Complexity threshold study on MedQA Full set and Hard set.

## F. IIE Metric Design Discussion

*Table 7.* Comparison of efficiency metrics on 5 hard sets. IIE$_{ZS}$ and IIE$_{CoT}$ denote IIE with Zero-Shot and CoT as baseline, respectively.

| Method | Acc/Cost | IIE$_{ZS}$ | IIE$_{CoT}$ |
|---|---|---|---|
| Zero-Shot | 35.303 | REF | – |
| CoT | 6.271 | 1.487 | REF |
| AFlow | 0.684 | 0.245 | 0.141 |
| MEDCOG-ALL | 1.005 | 0.336 | 0.181 |
| MEDCOG-META | **1.432** | **0.595** | **0.438** |

We discuss the design of Inference Incremental Efficiency (IIE) and compare it with alternative efficiency metrics in Table 7.

**Limitations of Accuracy/Cost:** Although Accuracy/Cost is a straightforward baseline-agnostic metric, it conflates intrinsic LLM capability with method-specific gains. First, a random guess on 4-choice questions already yields 25.0% accuracy, and fixed input overheads (question, context) are shared across all methods, so neither the numerator nor the denominator truly reflects incremental contribution. As a result, Accuracy/Cost systematically favors low-effort methods: Zero-Shot achieves the highest ratio (7.2) despite being the weakest baseline.

**Why CoT as IIE Baseline:** Inspired by the Incremental Cost-Effectiveness Ratio (ICER) in health economics (Weinstein & Stason, 1977), IIE measures the marginal accuracy gain per additional unit of cost relative to a standard-of-care baseline. We choose CoT for three reasons: (1) CoT is the standard reasoning paradigm adopted by all compared methods, ensuring a fair incremental comparison; (2) using Zero-Shot as the baseline would conflate the gains from CoT reasoning itself with those of the method, since all baselines except Zero/Few-Shot include CoT or its variants; (3) CoT represents the minimum necessary and reasonable expenditure for reasoning, and subtracting its cost cleanly isolates each method's additional contribution. As shown in Table 7, MEDCOG-META achieves the highest IIE under both Zero-Shot and CoT baselines, confirming robust marginal efficiency regardless of the anchor choice.

## G. Scaling Curve Sensitivity

We verify that the Inference Density conclusions are robust to the choice of fitting function. Beyond the logarithmic fit reported in the main text, we additionally fit the reference Pareto frontier with linear and power-law functions. Linear fitting yields lower $R^2$ since accuracy cannot grow linearly with cost. Power-law fitting achieves comparable $R^2$ but produces degenerate density estimates when the model's accuracy surpasses the function's asymptote, limiting its interpretability. MEDCOG-META maintains high Inference Density ($> 2\times$) across all fitting variants and across all 5 hard sets, confirming that the efficiency gains are not an artifact of a particular curve choice. We therefore adopt the logarithmic function for its balance of fit quality and stability.

## H. Calibration Stability

*Table 8.* Accuracy on MedQA-H under different calibration settings.

| Calibration | GPT-4o | GPT-4o-mini | o3-mini | Gemini-2.0-Flash | Qwen3-8B |
|---|---|---|---|---|---|
| w/o calibration | 52 | 40 | 44 | 48 | 46 |
| w/ calibration | **52** | **50** | **49** | **52** | **47** |
| w/ upper quartile | 47 | 46 | 42 | 46 | 46 |
| w/ median | 52 | 48 | 49 | 47 | 42 |
| w/ lower quartile | 48 | 47 | 49 | 48 | **52** |

Table 8 compares four calibration settings: (1) no calibration (directly applying GPT-4o thresholds to other backbones), (2) per-backbone calibration on 50 held-out samples, and statistical alternatives including (3) upper quartile, (4) median, and (5) lower quartile of each backbone's score distribution.

Per-backbone calibration consistently achieves the best or competitive performance across most backbones, confirming the effectiveness of lightweight calibration. Statistical alternatives (median, quartile) also largely outperform the no-calibration setting, demonstrating robustness to threshold choice within a reasonable range. A notable exception is Qwen3-8B, where the lower quartile threshold (52) outperforms per-backbone calibration (47), suggesting that Qwen3-8B's meta-cognitive scores are systematically underestimated, and a lower threshold better compensates for its conservative self-assessment tendency. Moreover, the calibration on valid set limits the threshold selection accuracy. These observations further highlight the importance of backbone-aware calibration for different LLMs.

## I. Case Study

### I.1. Full Strategy Case

**Case Analysis:** Table 13, 14, and 15 demonstrate a case using the combined strategy SCoT+KG+MEMORY to reason through a clinical scenario of right ureter and renal pelvis dilation. The SCoT systematically links symptoms, imaging, and anatomical risk factors to identify mechanical compression from a common iliac artery aneurysm as the likely cause, while the KG provides directional guidance by connecting vascular complications to ureteral obstruction, and Memory reinforces the reasoning with precedent cases.

**Post-Hoc Analysis:** We also conduct a Post-hoc evaluation with GPT-5.2. It shows that KG (0.7) correctly orients the reasoning but is limited by edges in KG, Memory (0.9) supplies clinically relevant examples, and SCoT (0.8) structures the multi-step derivation, demonstrating that MEDCOG effectively monitors and integrates multiple types of knowledge to solve complex clinical queries.

## I.2. Error Analysis

*Table 9.* Error Type Statistics. For 0/1/2 types, the statistic method is different for Strategy Pool and MedCoG-Meta, divided by "/".

| | Strategy Pool | MedCoG-Meta |
|---|---|---|
| 0. All strategies in strategy pool are correct / selected right strategy | 18 | 52 |
| 1. Feeling of Knowing: DA and SCoT wrong, SCoT+KG or SCoT+Mem or SCoT+KG+Mem correct / More knowledge strategy correct | 13 | 4 |
| 2. Over Reasoning: DA or SCoT correct, SCoT+Mem or SCoT+KG or SCoT+KG+Mem wrong / Less knowledge strategy correct | 33 | 14 |
| 3. Synergy Missed: SCoT+KG+Mem correct, SCoT+KG or SCoT+Mem wrong | 29 | 4 |
| 4. Memory Noise: SCoT correct, SCoT+Mem wrong; SCoT+KG correct, SCoT+KG+Mem wrong | 20 | 3 |
| 5. KG Noise: SCoT correct, SCoT+KG wrong; SCoT+Mem correct, SCoT+KG+Mem wrong | 23 | 10 |
| 6. Information Conflict: SCoT+KG correct and SCoT+Mem correct, SCoT+KG+Mem wrong | 5 | 2 |
| 7. Unsolvable: All strategies are wrong. | 33 | 33 |

To investigate the reliability of the meta-cognitive regulator, we classified failure cases on the MedQA Hard set into 7 types, as shown in Table 9. We have tallied the number of different types of error cases in (1) the strategy pool, (2) the strategy selected by MEDCOG-META.

**Negative Impacts Reduction:** The analysis demonstrates that MEDCOG-META significantly enhances knowledge synergy and mitigates the negative impacts of external information. First, meta-cognitive regulation effectively facilitates the complementarity of diverse knowledge sources; for instance, "Synergy Missed" errors were drastically reduced from a potential 29 to just 4, proving the regulator's precision in identifying complex scenarios requiring joint support from KG and Memory. Secondly, regarding negative information impacts, the system effectively filtered noise, suppressing "Memory Noise" from 20 potential cases to only 3 and "KG Noise" from 23 to 10.

**Insights:** There are also critical insights that emerge from the data: (1) Only 13 cases in the pool genuinely required external knowledge (Feeling of Knowing), suggesting that knowledge deficits are not the primary cause of difficulty for most hard samples; (2) "Over-Reasoning" still accounted for 14 errors, highlighting the critical importance of enabling Large Language Models to maintain higher confidence in their internal knowledge and avoid indiscriminate reliance on external retrieval; (3) KG brings relatively more noise, possibly attributed to the noise in structural paths and inherent knowledge coverage incompleteness.

# J. Effect of Memory Domain: In-Distribution vs. Out-of-Distribution

*Table 10.* The performance of PubMedQA with both in-distribution and out-of-distribution memory.

| PUBMEDQA | ID | OOD |
|---|---|---|
| MEDCOG W/O MEM | 12.0 | |
| MEDCOG-ALL | 13.0 | 19.0 |
| MEDCOG-META | 10.0 | 20.0 |

Since PubMedQA exploits the unique "yes/no/maybe" reasoning, rather than the multi-choice paradigm in other datasets, we explore both in-distribution and out-of-distribution (OOD) memory for PubMedQA, as shown in Table 10. We construct the in-distribution Case Bank with 2,350 PubMedQA SCoT experiences from MedReason (Wu et al., 2025a). The OOD Case Bank is the 2,707 SCoT cases of MedQA (1,402) and MedMCQA (1,305) from MedReason.

*Table 11.* Ablation Study. The GPT-4o is our backbone.

| STRATEGY | MEDQA-F | MEDQA-H | MEDMCQA-H | MMLU-H | MMLU-PRO-H | PUBMEDQA-H |
|---|---|---|---|---|---|---|
| ***Backbone LLMs*** | | | | | | |
| O3-MINI | 92.7 | 53.0 | 24.0 | 35.6 | 15.0 | 16.0 |
| GPT-4O | 87.8 | 32.0 | 25.0 | 24.7 | 21.0 | 9.0 |
| ***Meta-Cognition Strategies*** (GPT-4O) | | | | | | |
| ORACLE | **98.98** | **67.0** | **59.0** | **53.4** | **51.0** | **37.0** |
| META | **90.02** | **52.0** | **36.0** | **35.6** | **44.0** | **20.0** |
| ***Static Strategies*** (GPT-4O) | | | | | | |
| SCOT | 89.55 | 41.0 | 35.0 | **35.6** | 37.0 | 10.0 |
| SCOT+MEM | 89.08 | 42.0 | 30.0 | 32.9 | 41.0 | 12.0 |
| SCOT+KG | 87.43 | 37.0 | 28.0 | 30.1 | 38.0 | 10.0 |
| SCOT+KG+MEM | 88.85 | 50.0 | 32.0 | 28.8 | 36.0 | 19.0 |

**OOD Transferability via Diverse Patterns:** Despite the structural discrepancy between PubMedQA's "Yes/No/Maybe" format and the multi-choice nature of MedQA/MedMCQA, using OOD memory yields significantly higher performance (20.0% vs 10.0%). We attribute this to the richness of the OOD sources: the diverse and complex clinical reasoning patterns inherent in MedQA/MedMCQA provide more robust logical references than PubMedQA's own narrower trajectories, effectively guiding the model through analogical reasoning across domains.

**Meta-Cognitive Precision:** MEDCOG-META outperforms MEDCOG-ALL in the OOD setting (20.0% vs 19.0%), confirming that the regulator successfully utilizes the Familiarity score to filter relevant experiences. By selectively retrieving high-quality OOD analogs only when they align with the current problem's latent logic, the system avoids negative transfer from irrelevant cases.

**Limitations of ID Trajectories:** Conversely, In-Distribution performance is suboptimal, with MEDCOG-META falling behind the baseline. This suggests that the reasoning trajectories specific to the "Yes/No/Maybe" format are inherently sparse or difficult to generalize as episodic memory. Consequently, the meta-monitor may frequently perceive these ID cases as low-utility, suppressing memory activation, which—in this specific format—leads to under-performance compared to the static inclusion of all cases.

## K. Ablation Study

In this section, we provide a comprehensive ablation study to validate the effectiveness of each component in MEDCOG and analyze the impact of different knowledge sources. The detailed results are presented in Table 11.

**Impact of Meta-Cognition Regulation:** The comparison between META and other static strategies reveals the critical role of dynamic on-demand reasoning. On multiple datasets, the static inclusion of Knowledge Graphs (SCOT+KG) leads to a performance drop compared to the baseline SCOT, attributed to the retrieval of irrelevant or noisy paths. However, MEDCOG-META achieves better overall performance, effectively activating KG only when necessary, thereby mitigating negative transfer. Moreover, for MedMCQA, MMLU, and MMLU-Pro, SCOT with decoupled structural path and reasoning trajectories has already achieved competitive results, demonstrating the necessity of meta-cognition regulation for preventing over-reasoning and noise involvement.

**Synergy Effects between KG and Memory:** A notable phenomenon is observed on MedQA-F, MedQA-H, and PubMedQA-H. While SCOT+KG performs poorly on these datasets, their combination with MEMORY, i.e., SCOT+KG+MEM, achieve significant boost on these datasets. This implies a strong synergy effect where episodic memory provides contextual examples that help the LLM interpret and ground the abstract structural paths retrieved from the Knowledge Graph.

**The Oracle Gap:** The ORACLE row represents the theoretical upper bound where the optimal strategy is always selected. On all datasets, ORACLE shows significantly higher performance than all strategies. This substantial gap indicates that while MEDCOG's regulator is effective, the underlying LLM (GPT-4o) possesses the potential to solve even more complex cases if the meta-cognitive calibration can be further refined to perfectly match the problem state with the reasoning strategy.

# L. Memory Evolving

*Table 12.* The performance of Case Bank construction on the MedQA Train set in round 1 and round 2. In each round, the LLM runs the train set samples through SCOT+KG+MEMORY strategy.

| ROUND 1 (W/ 50 SEED MEMORY) | ROUND 2 (W/ ROUND 1 AS MEMORY) |
|---|---|
| 80.35 | 82.19 |

While our main experiments utilize the high-quality, filtered SCoT data from MedReason (Wu et al., 2025a) to construct the Case Bank, we further investigate whether MEDCOG possesses the capability to construct and evolve its episodic memory from scratch, independent of external datasets.

**Experimental Settings:** We first construct a seed memory case bank with 50 samples from SCOT+KG. In round 1, we run samples with SCOT+KG+MEMORY strategy, in which the memory is seed memory. In round 2, the samples are also conducted with SCOT+KG+MEMORY strategy, in which the memory is the round 1 SCoT trajectories.

**Evolving Memory Dynamics:** As shown in Table 12, the Case Bank successfully evolves in round 2, where episodic knowledge from round 1 successfully allows the LLM to solve more cases in round 2. It indicates that the quality of self-generated memory is able to evolve by learning from its own historical successful and failed experiences, validating the evolving dynamics of episodic memory in medical reasoning.

**Insights:** Given our primary focus on the meta-cognitive regulation capabilities of LLMs, we directly leverage SCoT trajectories from MedReason, filtered via their knowledge anchoring and quality control, and yielding a Case Bank composed of successful experiences. Thanks to their efforts, we are able to construct a cost-effective Case Bank, eliminating the computational costs of constructing a case bank from scratch. The continuous evolution of medical agents, particularly regarding their meta-cognitive regulation dynamics, would be worthy of interest for future exploration.

# M. Limitations and Future Work

**Medical Domain:** Although the proposed MEDCOG framework is theoretically agnostic to the reasoning domain, this study strictly prioritizes **medical reasoning**. We focus on this domain due to its distinct demands for high-precision decision-making, handling of noisy clinical contexts, and dependency on specialized, multi-hop knowledge. Consequently, the generalization of MEDCOG to open-domain benchmarks remains a subject for future exploration.

**Meta-Cognition Calibration:** Regarding the meta-cognitive planning phase, we currently employ a heuristic approach where thresholds are empirically determined via hyper-parameter search on a held-out validation set. This way, the intrinsic uncertainty and meta-cognitive assessments of LLMs are leveraged and bring transparency to the knowledge regulation process. Despite its simplicity, the regulator already yields consistent gains across datasets and strategies. More importantly, it provides an intuitive peek at the LLM meta-cognition through meta-cognitive scores distributions, strategy distributions, and threshold study, and allows us to observe the limitations of LLM meta-cognition, providing insights for future study. The limitation lies in the assumption of distributional consistency between validation and test data. In real-world deployments, significant distribution shifts (e.g., varying difficulty levels across patient cohorts) could potentially impact the precision of these static thresholds. Future work could explore adaptive meta-cognition calibration or learnable policy networks to dynamically adjust regulation sensitivity for robust out-of-distribution generalization.

**Knowledge Completeness:** The effectiveness of the KG and Memory strategies is inevitably bound by the coverage and quality of the knowledge graph and case bank. Regarding *factual knowledge*, we observed inherent sparsity in Knowledge Graphs where certain verification-hypothesis pairs lacked bridging paths, limiting the retrieval of evidence for specific long-tail entities. Similarly, for *episodic knowledge*, its finite scale may not encompass every reasoning pattern, especially for rare clinical scenarios. While MEDCOG mitigates these gaps by dynamically balancing external KG/Memory retrieval with intrinsic parametric knowledge, it does not actively repair the underlying sources. Encouragingly, our empirical observations demonstrate the feasibility of evolving episodic memory in Appendix L. This suggests that medical memory can adaptively grow and refine for life-long medical learning. This way, rare reasoning patterns can be involved, and outdated cases can be edited or deleted. Thus, future research in developing automated knowledge evolution mechanisms, including factual and episodic knowledge, would be pivotal for lifelong medical reasoning and agents.

Moreover, a multi-agent framework with meta-cognitions from different perspectives, episodic knowledge from knowledge

graph verification plans, would be promising for future explorations.

## N. Prompts

The prompts for our MEDCOG framework are listed in Listing 1 (Monitoring), Listing 2 (Planning), Listing 3 (Evaluating), and Listing 4 (Structural CoT). For medical entity grounding procedures, we referred to the prompts in MedReason (Wu et al., 2025a) and adapted them to our verification query and hypothesis pairs and question-path similarity-based path ranking.

```
1  **YOUR ROLE:**
2  You are a **Metacognitive Monitor** for a medical AI. Your job is to ASSESS the problem,
   ↪ not answer it.
3  You must evaluate the incoming question based on three dimensions to decide the reasoning
   ↪ strategy.
4
5  **INPUT:**
6  - Question and Options.
7
8  **EVALUATION CRITERIA:**
9  1. **Complexity (0-1):** Does it require multi-hop reasoning or logic? (1=High complexity)
10 2. **Familiarity (0-1):** Is it a standard clinical case of common knowledge or a rare
   ↪ clinical case of a hard reasoning pattern? (1=High familiarity)
11 3. **Knowledge-Density (0-1):** Does it rely on specific, obscure medical knowledge,
   ↪ including medical concepts or entities (genes, rare drugs)? (1=High density)
12
13 **STRATEGY SELECTION:**
14 - **DIRECT:** If Complexity < 0.5. (Simple/General knowledge)
15 - **MEMORY_RECALL:** If Familiarity > 0.5. (Standard but complex cases, use similar past
   ↪ cases/CoT)
16 - **KNOWLEDGE_GRAPH_VERIFICATION:** If Knowledge-Density > 0.5. (Requires external
   ↪ knowledge graph verification)
17
18 **OUTPUT FORMAT:**
19 Reply ONLY in JSON:
20 { \"complexity\": FLOAT, \"familiarity\": FLOAT, \"knowledge_density\": FLOAT, \"reasoning
   ↪ \": "Brief␣explanation␣of␣why␣these␣scores␣are␣given." }
```

*Listing 1.* Prompt for Monitoring of Meta-Cognition Regulator

```
1  **YOUR ROLE:**
2  You must formulate a **verification strategy** to identify the correct option among the
   ↪ candidates. You have access to a Knowledge Graph (KG) via an EXECUTOR to verify
   ↪ connectivity and relationships between the Question and your hypothesis.
3
4  **INPUT:**
5  - A high-level medical question.
6  - A list of candidate options (e.g., A, B, C, D).
7  - Retrieved Experience Examples: Similar past cases containing both positive and negative
   ↪ reasoning patterns.
8
9  **YOUR TASK:**
10 - Analyze the problem contexts and formulate specific hypotheses for verification; the
   ↪ EXECUTOR will then explore the Knowledge Graph and return the structural concept
   ↪ paths for verification.
11 - You may target candidate options directly or decompose the reasoning path into
   ↪ intermediate sub-steps.
12 - Output the target questions and hypotheses ONLY in JSON with the schema:
13 { \"plan\": [ {\"id\": INT, \"question\": STRING, \"hypothesis\": STRING} ...] }
14 - After each task is executed by the EXECUTOR, you will receive its result. Please
   ↪ carefully consider the specific clinical profiles and recall relevant knowledge to
   ↪ solve the problem, and take these factors into account when planning and giving the
   ↪  final answer.
15 - Keep the hypotheses minimal, and retain only those strictly relevant to inferring the
   ↪ correct answer from the given options.
```

```
16  - After you receive results from EXECUTOR, you must REFINE your plan if you need more
       ↪ evidence or other knowledge, and emit a *new* JSON plan for the remaining work.
17  - If you have enough knowledge OR reaching the final reasoning step, STOP generating json
       ↪ style plan. you must integrate your internal parametric knowledge with the
       ↪ retrieved external knowledge to deduce the conclusion.
18  Your final output must strictly follow the format below:
19
20  ### Finding reasoning paths:
21  (List the key logical hops found in the KG or inferred from your internal knowledge, e.g.
22  1. [Key concept from Question] -> [Relation] -> [Intermediate concept] -> [Key entity from
       ↪  Option]
23  2. [Key observation from Question] -> [Intermediate concept] -> [Key conclusion from
       ↪ Option])
24
25  ### Reasoning Process:
26  (Generate a step-by-step reasoning process to solve the problem. Ensure the steps are
       ↪ logical and concise.)
27
28  ### FINAL ANSWER: [Option Letter]
29
30  **CONSTRAINTS:**
31  - Your final answer should be the Option Letter only, e.g., A, B, .... Do not provide
       ↪ extra explanation.
32  - Keep cycles as few as possible.
33  - Reply with *pure JSON only* for the planning phase.
34  - If you have enough knowledge OR reaching the final reasoning step, STOP generating json
       ↪ style plan. You must integrate your internal parametric knowledge with the
       ↪ retrieved external knowledge to generate the reasoning process to conclude the
       ↪ answer.
```

*Listing 2.* Prompt for Planning of Meta-Cognition Regulator

```
1   You are a strictly critical judge evaluating a retrieval process for a medical question.
2
3   --- USER QUERY ---
4   {query}
5
6   --- PLAN EXECUTED and RETRIEVED KNOWLEDGE PATHS ---
7   {results_text}
8
9   --- YOUR TASK ---
10  1. Assess if the retrieved paths provide a COMPLETE and LOGICAL answer to the query.
11  2. Identify specific gaps (e.g., missing drug side effects, broken relationship between A
       ↪ and B).
12  3. If information is missing, provide a SPECIFIC instruction for the next planning
       ↪ question and hypothesis step.
13
14  Output format strictly as JSON:
15  {{
16      "sufficient": boolean,
17      "reasoning": "short explanation",
18      "feedback_for_planner": "Direct instruction on what to question and hypothesis for
          ↪ next plan. E.g., 'The path mentions X but not its dosage. Search for dosage of
          ↪ X.'"
19  }}
```

*Listing 3.* Prompt for Evaluating of Meta-Cognition Regulator

```
1   You are a medical expert. You need to answer the medical question by choosing from one of
       ↪ the answer choices.
2   Please carefully consider the specific clinical profiles and recall relevant knowledge to
       ↪ solve the problem, and take these factors into account when planning and giving the
       ↪  final answer.
3   Your final output must strictly follow the format below:
```

```
4  ### Finding reasoning paths:
5  (List the key logical hops found in the KG or inferred from your internal knowledge, e.g.
6  1. [Key concept from Question] -> [Relation] -> [Intermediate concept] -> [Key entity from
       ↪  Option]
7  2. [Key observation from Question] -> [Intermediate concept] -> [Key conclusion from
       ↪ Option])
8
9  ### Reasoning Process:
10 (Generate a step-by-step reasoning process to solve the problem. Ensure the steps are
       ↪ logical and concise.)
11
12 ### FINAL ANSWER: [Option Letter]
13
14 **CONSTRAINTS:**
15 - Your final answer should be the Option Letter only, e.g., A, B, .... Do not provide
       ↪ extra explanation.
```

*Listing 4.* Prompt for Structural Chain-of-Thought

*Table 13.* Case Study (Part I).

| | | |
|---|---|---|
| **Question** | | A 68-year-old male comes to the physician for evaluation of right flank pain. He has a history of diabetes and peripheral artery disease. His blood pressure is 160/90 mm Hg. Physical examination shows abdominal tenderness and right flank tenderness. An ultrasound shows dilation of the right ureter and renal pelvis. Which of the following is the most likely underlying cause of this patient's condition?
A. Renal artery stenosis
B. Benign prostatic hyperplasia
C. Common iliac artery aneurysm
D. Urethral stricture |
| **Monitoring** | | "complexity": 0.6,
"familiarity": 0.8,
"knowledge_density": 0.7,
"reasoning": "The complexity is moderate because the case requires the integration of ultrasound findings with patient history to deduce the cause of the problem. Familiarity is high as the question relates to common conditions encountered in clinical practice, such as renal artery stenosis and benign prostatic hyperplasia, which are well-known causes of hydronephrosis. Knowledge-density is high because it involves specific medical conditions that affect the urinary tract anatomy and function, which require certain expertise to understand the implications of the ultrasound findings." |
| **Strategy** | | SCoT+KG+Memory |
| **Factual Knowledge** | **Plan** | "id" : 1,
"question" : "What condition leads to right ureter and renal pelvis dilation as observed in the ultrasound?",
"hypothesis" : "Common iliac artery aneurysm causes compression leading to ureter obstruction and hydronephrosis."

"id" : 2,
"question" : "What are the vascular complications associated with peripheral artery disease that may affect the ureter?",
"hypothesis" : "Peripheral artery disease can lead to vascular alterations resulting in external compression, such as from an iliac artery aneurysm." |
| | **Path** | Knowledge Path for Task 1: 1: right ureter → ureter → lateral structure → adrenal gland → dstyk → ureteropelvic junction obstruction
2: right ureter → ureter → lateral structure → kidney → dstyk → ureteropelvic junction obstruction
3: right ureter → ureter → lateral structure → lung → dstyk → ureteropelvic junction obstruction
4: renal pelvis → anatomical conduit → multicellular anatomical structure → tissue → connective tissue → dstyk → ureteropelvic junction obstruction
5: renal pelvis → anatomical conduit → multicellular anatomical structure → multicellular organism → embryo → dstyk → ureteropelvic junction obstruction

Knowledge Path for Task 2: 1: ureter → lateral structure → kidney → adra2b → carvedilol → peripheral arterial disease
2: peripheral arterial disease → arsenic → pptc7 → kidney → lateral structure → ureter
3: ureter → lateral structure → kidney → usp34 → vehicle emissions → peripheral arterial disease
4: ureter → lateral structure → kidney → c9orf64 → coronary artery → systemic artery → iliac artery
5: ureter → lateral structure → kidney → mapre3 → coronary artery → systemic artery → iliac artery |

*Table 14.* Case Study (Part II).

| | |
|---|---|
| **Episodic Knowledge** | Positive Examples (reward=1) - Showing 5 of 5:
Example 1:
Question: A 42-year-old woman comes to the physician because of right flank pain that started 3 days following a procedure. Her vital signs are within normal limits. Physical examination shows right costovertebral angle tenderness. An intravenous pyelogram shows a dilated renal pelvis and ureter on the right with a lack of contrast proximal to the ureterovesical junction. This patient most likely recently underwent which of the following procedures?
Final Output:

Finding reasoning paths:
1. Flank pain → Recent surgical history → Possible complication → Ureteral obstruction → Right flank pain.
2. Flank pain → Costovertebral angle tenderness → Hydronephrosis → Ureteral obstruction → Right flank pain.
3. Intravenous pyelogram findings → Dilated renal pelvis and ureter → Obstruction at ureterovesical junction → Possible surgical or anatomical cause.
4. Ureteral anatomy → Proximity to pelvic structures → Surgical procedure (e.g., hysterectomy) → Ureteral injury → Right flank pain.

Reasoning Process:
1. **Symptoms and Physical Exam Findings**: - The patient presents with right flank pain and costovertebral angle tenderness, which are classic signs of renal or ureteral pathology. These symptoms suggest a problem involving the kidney or ureter, such as obstruction or infection.

2. **Imaging Findings**: - The intravenous pyelogram shows a dilated renal pelvis and ureter on the right side, which is consistent with hydronephrosis. The lack of contrast proximal to the ureterovesical junction indicates a blockage at this location.

3. **Possible Causes of Ureteral Obstruction**: - Ureteral obstruction can result from intrinsic causes (e.g., stones, strictures, tumors) or extrinsic causes (e.g., compression or injury from nearby structures). Given the recent onset of symptoms following a procedure, a surgical complication is a strong possibility.

4. **Surgical History and Anatomical Considerations**: - The ureter runs close to the uterus and can be injured during pelvic surgeries, such as a hysterectomy. Ureteral injury during a hysterectomy is a known complication, often due to accidental ligation, kinking, or cutting of the ureter. This could explain the obstruction at the ureterovesical junction and the resulting hydronephrosis.

5. **Correlation with the Patient's Presentation**: - The timing of the symptoms (3 days after a procedure) aligns with the development of hydronephrosis due to ureteral obstruction. The right-sided findings suggest that the procedure likely involved manipulation or injury to the right ureter. |

*Table 15.* Case Study (Part III).

| | |
|---|---|
| **Procedural Knowledge** | Finding reasoning paths:
1. Right flank pain and tenderness → Ultrasound shows right ureter and renal pelvis dilation → Possible obstruction → Common iliac artery aneurysm causing compression.
2. Peripheral artery disease → Vascular alterations → Potential for iliac artery aneurysm → Compression of the ureter leading to obstruction and hydronephrosis.

Reasoning Process:
1. **Symptom Analysis**: The patient, a 68-year-old male with diabetes and peripheral artery disease, presents with right flank pain, abdominal tenderness, and right flank tenderness. These symptoms suggest a possible obstruction or compressive cause in the urinary tract.

2. **Imaging and Physical Examination**: The ultrasound showing dilation of the right ureter and renal pelvis suggests hydronephrosis due to a possible obstruction at or below the level of the ureter. This aligns with the findings of abdominal and flank tenderness, indicating possible urinary tract blockage.

3. **Underlying Causes**: - **Peripheral Artery Disease Influence**: The patient's history of peripheral artery disease can predispose him to vascular complications such as aneurysms. This is significant because an aneurysm in the common iliac artery can exert pressure on nearby structures, such as the ureter, causing obstruction. - **Path Verification**: The knowledge paths suggest that the anatomical positioning of the iliac artery can affect the ureter due to its proximity, especially if a vascular anomaly like an aneurysm occurs.

4. **Exclusion of Other Options**: - Renal artery stenosis typically presents with hypertension and not flank pain. - Benign prostatic hyperplasia would primarily cause lower urinary tract symptoms rather than isolated ureteral dilation. - Urethral stricture is more likely to cause bladder-level or bilateral kidney problems rather than unilateral.

5. **Connecting Evidence**: The plausible connection of peripheral artery disease leading to complications in iliac artery integrity supports the hypothesis that a common iliac artery aneurysm is the most likely cause of the observed hydronephrosis.

FINAL ANSWER: C |
| **Post-hoc Evaluation (GPT-5.2)** | ***KG Evaluation***
**Score** (Assign a score between 0 and 1 reflecting the effectiveness of the strategy.): 0.7
**Justification** (Provide a concise but precise justification explaining the reasons and limitations): The knowledge graph–based plan correctly identified the reasoning direction by linking ureteral obstruction with iliac vasculature and peripheral artery disease, but its effectiveness was limited because the KG did not explicitly encode the aneurysm-related enlargement and mechanical compression mechanism, reflecting a structural limitation rather than an incorrect plan.

***Memory Evaluation***
**Score**: 0.9
**Justification**: The memory example is effective because it provides a complete, clinically grounded causal chain that transparently connects symptoms, imaging findings, anatomy, and context, making the reasoning reliable and reusable across similar cases.

***SCoT Evaluation***
**Score**: 0.8
**Justification**: The structural chain-of-thought is effective because it systematically integrates symptoms, imaging, risk factors, and anatomical relationships to narrow the hypothesis space, even though the critical aneurysm compression step relies on implicit medical knowledge. |

