# OpenReview forum: "MedCoG: Maximizing LLM Inference Density in Medical Reasoning via Meta-Cognitive Regulation"
_ICML.cc/2026/Conference — ICML 2026 regular_

### Official Review · Reviewer_pM5Z · 2026-03-12

**Soundness:** 3
**Presentation:** 4
**Significance:** 3
**Originality:** 3
**Overall Recommendation:** 4
**Confidence:** 4

**Summary:**

This paper presents MEDCoG (Medical Meta-Cognition Agent with Knowledge Graph), a medical reasoning agent that uses meta-cognitive self-assessment to route LLM inference across three knowledge types: procedural knowledge via SCoT, episodic knowledge via memory, and factual knowledge via KG retrieval. The routing policy is governed by three monitored dimensions (complexity, familiarity, and knowledge density) implemented through non-parametric thresholds. Across hard subsets of five medical benchmarks, the paper argues that selective on-demand reasoning outperforms both static augmentation and several agentic baselines, surpassing AFlow by 8.7% in average accuracy, and introduces inference density (yielding a 5.5x gain) and inference incremental efficiency as metrics for reasoning efficiency.

**Compliance With Llm Reviewing Policy:**

Affirmed.

**Final Justification:**

I thank the authors for the detailed rebuttal and additional experiments. The sensitivity analysis on threshold robustness, the stronger results relative to prior methods, and the Precision/Recall/F1 breakdown of the meta-cognitive router substantially mitigate my earlier concerns about the heuristic nature and stability of the framework. The clarifications around the intervention-efficiency metrics also make the evaluation setup and practical implications much clearer. My primary technical concerns have been substantially addressed, though the backbone sensitivity revealed by Qwen3-8B and the small hard-subset sizes leave some residual uncertainty about generalization.

**Key Questions For Authors:**

1. The meta-cognitive scores are produced by the same backbone LLM being regulated. Can the authors quantify how often the monitor’s strategy choice matches the Oracle-selected strategy, and how this routing accuracy varies across backbone models?

2. The three thresholds are calibrated on only 50 held-out samples. How sensitive is downstream accuracy to threshold perturbation, and how stable is performance under distribution shift between calibration and test settings?

3. The Oracle gap remains substantial on MedQA-H and several other datasets. Can the authors decompose how much of this gap is due to routing errors versus limitations of the available strategy pool itself?

4. IIE is defined relative to CoT as a fixed reference. Can the authors justify why CoT is the right anchor across all datasets, or provide a more baseline-agnostic efficiency analysis?

**Limitations:**

Mostly yes. The paper is reasonably candid about threshold sensitivity, knowledge sparsity, and domain scope. However, the self-referential calibration issue deserves stronger treatment, particularly because the monitor’s reliability is bounded by the same backbone whose behavior it is trying to regulate, and the backbone-dependent failure modes appear to be one of the more general findings of the paper.

**Strengths And Weaknesses:**

Soundness: The paper’s main strength is that it identifies and studies a real empirical phenomenon: indiscriminate knowledge augmentation can hurt performance, while selective routing recovers much of the benefit of a richer strategy pool. The pilot analysis in Table 1 is especially effective because it directly motivates the method by showing that some augmented strategies underperform simpler ones, while the Oracle upper bound is substantially higher. The inference density framing is also useful, and the empirical scaling-curve analysis provides a concrete way to discuss efficiency rather than accuracy alone. The main limitation is that the routing mechanism remains heuristic: the policy is a non-parametric gate with three thresholds calibrated on only 50 held-out samples, and the monitor scores are produced by the same backbone model being regulated. While the paper acknowledges this self-referential calibration problem and includes threshold studies, it does not fully establish how robust the routing policy is to threshold perturbation, backbone choice, or distribution shift, and the remaining Oracle gap is still substantial.

Presentation: The paper is dense but generally well organized, and the Monitor–Plan–Execute–Evaluate framework is clearly described. The figures are informative, especially the cost-accuracy and strategy-distribution analyses. One weakness is that one of the paper’s most interesting findings, namely the backbone-dependent meta-cognition behavior, is somewhat underemphasized relative to the main system results. A second issue is that the IIE metric depends on CoT as a fixed baseline, and that design choice could be better justified in the main text.

Significance: This is the paper’s strongest dimension. The paper addresses an important question in test-time scaling for medical reasoning: whether the main bottleneck is adding more computation or selecting the right reasoning strategy for each case. The reported inference density gains are striking, and the broader framing around selective reasoning is likely to be useful beyond the specific MEDCoG system. That said, the broader significance still depends on whether the routing behavior remains stable beyond the current benchmark setting and backbone configuration.

Originality: The individual components, including SCoT, KG retrieval, memory augmentation, and threshold-based routing, are established. The contribution is integrative: a training-free, instance-wise meta-cognitive routing framework over heterogeneous knowledge types, together with an efficiency framing based on inference density and IIE. Under ICML’s broader definition of originality, this is a meaningful contribution.

---

> ### Author Rebuttal · Authors · 2026-03-31
>
> We thank Reviewer pM5Z for the positive assessment and helpful suggestions. We will revise the manuscript as advised.
>
> >Soundness, Significance and Q2: Stability and Robustness of calibration method.
>
> Robustness is established across thresholds, backbones, distribution shifts, and wider benchmarks:
> - Threshold perturbation: On MedQA-H, when $\tau_f<=0.6$ and $\tau_k<=0.7$, accuracy is always above MedCoG-ALL (50) and AFlow (48), demonstrating stability inside certain intervals.
> - Backbones: Through per-backbone calibration, most LLMs outperform AFLow (48) as shown in **response to Q1 below**, demonstrating robustness.
> - Distribution shift: The distribution shift between the valid and test naturally exists. The effectiveness of MedCoG-Meta w/ calibration demonstrates robustness to such shifts.
> - Benchmarks: MedCoG-Meta achieves the highest scores on open-ended reasoning (MedQA-H-Open: 0.34) and multi-turn agent tasks (AgentClinic: 0.73), demonstrating benchmark generalization.
> - Low cost median calibrations also show stable performance across datasets (34.6), prompts (51.6 on MedQA-H), detailed in **R35U Q1**.
>
> >Presentation of Underemphasized findings.
>
> Thanks for pointing out the significance of our findings. We will summarize and highlight our findings, including this one in the Appendix.
>
> >Presentation & Q4: IIE based on CoT
>
> We added two metrics to illustrate the rigor in IIE based on CoT:
> - Accuracy/Cost efficiency, which is baseline-agnostic efficiency.
> - IIE based on Zero-Shot.
>
> ||acc/cost|IIE zero-shot|IIE CoT|
> |-|-|-|-|
> |Zero-Shot|7.2|REF|-|
> |CoT|6.0|3.6|REF|
> |AFlow|0.6|0.2|0.1|
> |MedCoG-ALL|1.0|0.4|0.2|
> |MedCoG-META|1.4|0.7|0.4|
>
> The full metric table is in https://anonymous.4open.science/r/MedCoG-anonymous-4555/efficiency_metric.md
>
> 1. Why is Accuracy/Cost wrong?
> - The basic ability is not isolated. The baseline spot is not $(0,0)$ since random guess for 4-choice questions will yield 25.0 accuracy. Moreover, the fixed input overheads (patient information, question, etc) are not isolated.
> - It results in unbalanced growing speed in the Accuracy and Cost, and favors methods with less effort, i.e., Zero-Shot achieves the best Acc/Cost.
>
> 2. Inspired by the medical economic metric ICER, we propose to measure the **intervention efficiency** compared to the "Standard of Care" ability of LLMs, presenting $IIE=\Delta acc / \Delta cost$.
>
> 3. Why is CoT better than Zero-Shot for IIE baseline?
> - CoT is chosen as IIE baseline as standard baseline representing the LLM basic ability. CoT is widely adopted in LLM reasoning studies: *all our baselines except Zero/Few-Shot include CoT or its variants*.
> - If the Zero-Shot is baseline, it would confuse whether the gains are from CoT or baseline contributions, e.g., the $\Delta Acc$ of AFlow will include the gains from CoT, and cannot truly reveal their contributions.
>
> >Originality:
>
> We sincerely thank you for pointing out the originality and its alignment to ICML definition. We are greatly encouraged.
>
> >Q1: Monitor-Oracle Matching Quantification
>
> 1. The global accuracy of Regulator to Oracle-chosen strategy is $52/67=77.61%$ on MedQA-H, where Regulator selected the Oracle strategy, while the regret is $1-77.61=22.39%$ due to suboptimal routing.
>
> 2. We conduct Precision/Recall/F1 analysis of the separated three dimensions. The tie situations, i.e., both right or both wrong, are removed from calculation.
>
> ||Meta|Complexity|Familiarity|Knowledge density|
> |-|-|-|-|-|
> |GPT-4o|52|0.95/0.63/0.75|0.83/1.00/0.90|0.59/0.95/0.80|
> |GPT-4o-mini|50|1.00/0.78/0.88|0.83/0.83/0.83|0.62/0.95/0.75|
> |Gemini-2.0-Flash|52|0.97/1.00/0.98|0.88/0.83/0.86|0.69/0.95/0.80|
> |o3-mini|49|0.97/0.88/0.92|0.65/1.00/0.79|0.68/0.72/0.70|
> |Qwen3-8B|47|0.97/0.94/0.95|0.47/0.64/0.54|0.75/0.21/0.33|
>
> Observations:
> - LLMs basically show above 50% Precision/Recall/F1, demonstrating meta-cognitive assessment ability. The metric on Complexity and Familiarity is higher than Knowledge.
> - o3-mini and Qwen3-8B show poorer alignment due to (1) less precise familiarity and knowledge assessments, (2) less coverage of all situations that require KG and memory.
> - GPT-4o, 4o-mini, and Gemini-2.0-Flash show less precision in the knowledge density assessments, which means more samples where KG brings noise are included.
>
> >Soundness & Q3: The Oracle gap.
>
> - Gap Calculation: The routing errors are the gap between Oracle (67) and Meta (52), i.e., $15$ samples on MedQA-H. The capability limitations of strategy pools are those between Oracle and 100, i.e., $33$ samples on MedQA-H.
> - Detailed Routing Error analysis: We detailed the routing errors in **Table 6**, where the errors are categorized into Meta-cognitive errors, Noise errors, and capability bound.
> - Oracle Gap improvements: MedCoG closes $15.8$%, $(37.5-34.5)/(53.5-34.5)$, gap between SOTA  (AFlow, 34.5) and Oracle (53.5) on 5 datasets. It demonstrates that our direction is feasible and reasonable, and points out a promising direction for future studies.

---

> > ### Author Rebuttal · Reviewer_pM5Z · 2026-04-02
> >
> > I thank the authors for the detailed rebuttal and additional experiments. The sensitivity analysis on threshold robustness, the stronger results relative to prior methods, and the Precision/Recall/F1 breakdown of the meta-cognitive router substantially mitigate my earlier concerns about the heuristic nature and stability of the framework. The clarifications around the intervention-efficiency metrics also make the evaluation setup and practical implications much clearer. My primary technical concerns have been substantially addressed, though the backbone sensitivity revealed by Qwen3-8B and the small hard-subset sizes leave some residual uncertainty about generalization.

---

> > > ### Author Response · Authors · 2026-04-03
> > >
> > > We thank the reviewer for the positive assessment. We would like to clarify the remaining generalization concerns as follows:
> > >
> > > >backbone sensitivity revealed by Qwen3-8B
> > >
> > > We agree with your observation regarding the smaller model. We further explore the phenomenon by experimenting on the Qwen series LLMs with larger sizes as follows:
> > >
> > > ||Meta|Complexity|Familiarity|Knowledge Density|
> > > |-|-|-|-|-|
> > > |GPT-4o|0.52|0.95/0.63/0.75|0.83/1.00/0.90|0.59/0.95/0.80|
> > > |Qwen3-8B|0.47|0.97/0.94/0.95|0.47/0.64/0.54|0.75/0.21/0.33|
> > > |Qwen3-32B|0.49|0.97/0.91/0.94|0.75/0.90/0.82|0.59/1.00/0.74|
> > > |Qwen3-Max|0.51|0.97/0.97/0.97|0.78/0.78/0.78|0.63/0.94/0.75|
> > >
> > > 1. **Limitations of Smaller LLMs**: As shown in the above table, Qwen3 with 32B size and Max version achieve better performance than AFlow and show higher precision and recall than Qwen3-8B in most dimensions. It demonstrates that Qwen3-8B is mainly limited by its parameter size, and **larger LLMs show better meta-cognitive regulation ability than smaller ones**. This limitation stems from the model's parameter scale rather than the framework design.
> > >
> > > 2. **Experimental Motivation**: Our primary intent in including Qwen3-8B was to provide comprehensive comparisons across LLM series and scales. Additional training may further elicit smaller LLMs' meta-cognitive ability, which we consider a promising future direction.
> > >
> > > 3. **Consistent Conclusions**: Our main conclusions hold for **large-scale, API-based LLMs: they consistently outperform SOTA AFlow and reduce cost through adaptive regulation**. It demonstrates that meta-cognitive regulation generalizes across LLM families and scales.
> > >
> > > >the small hard-subset sizes
> > >
> > > We address this concern from the following perspectives:
> > >
> > > 1. **Task Difficulty**: The hard sets contain samples where more than 50% LLMs fail. We focus on the challenging subsets to explore the actual contributions of MedCoG-Meta and the advanced baselines.
> > >
> > > 2. **Task Diversity**: The five hard sets cover distinct and common medical reasoning types. As detailed in **Table 5**, these datasets cover **3 reasoning formats** (4-choice/10-choice/yes-or-no), and **various reasoning domains** (clinical diagnosis/medical knowledge/medical context understanding). **The 5 hard sets, as well as the open-ended and multi-turn tasks, further establish our generalization across benchmarks.**
> > >
> > > 3. **Full-set Evaluations**: We also evaluated MedCoG-Meta on MedQA Full set, which includes more diverse difficulty levels. MedCoG-Meta remains consistently effective on the full set (accuracy 90.02), detailed in **Table 8**, further verifying our robustness and difficulty generalization.
> > >
> > > 4. **Fundamental Generalization of MedCoG-Meta**: MedCoG-Meta adaptively assesses each question to trigger the required strategies. It dynamically handles different difficulty levels, common/rare cases, and diverse knowledge granularity, achieving generalization to different benchmarks by design.

---

### Official Review · Reviewer_sXmp · 2026-03-12

**Soundness:** 2
**Presentation:** 3
**Significance:** 3
**Originality:** 2
**Overall Recommendation:** 4
**Confidence:** 4

**Summary:**

This paper introduces **MedCoG**, a novel framework that applies meta-cognitive regulation to optimize LLM inference efficiency in medical reasoning tasks. The core contribution lies in dynamically selecting inference strategies based on LLM's self-assessment of problem complexity, familiarity, and knowledge density. The framework integrates three types of knowledge: procedural knowledge (SCoT), factual knowledge (knowledge graphs), and episodic knowledge (memories). The authors report achieving **5.5× Inference Density** and **0.443 IIE** (Inference Increment Efficiency), demonstrating significant improvements over baseline methods on medical reasoning benchmarks.

**Compliance With Llm Reviewing Policy:**

Affirmed.

**Final Justification:**

Author's rebuttal addresses my concerns.

**Key Questions For Authors:**

1. **What exact routing rule was used in the experiments?**
Please answer this precisely, because Equation (1), the explanatory text below it, and Section 3.3 currently disagree. If you can provide the exact implemented logic and show that the paper’s results correspond to the intended rule, that would materially increase my confidence.

2. **How robust are the key gains in Table 2 under statistical testing?**
Since several gains are only one to a few questions on 73- or 100-example subsets, can you provide paired significance tests or bootstrap intervals for MEDCOG-META versus MEDCOG-ALL, SCoT, and AFlow? If the gains are not robust, that would weaken the paper substantially.

3. **How much of the claimed efficiency improvement survives under same-harness reruns and latency accounting?**
The cost-sensitive comparisons currently mix imported baseline numbers with new results. Can you provide apples-to-apples token and latency comparisons for at least a few strongest baselines? A clear answer here could move my score upward.

4. **How accurate is the regulator relative to oracle strategy choice?**
Since MEDCOG-ORACLE is central to the motivation, can you report the regulator’s agreement with oracle or the regret from suboptimal routing? If the regulator is often wrong, then the main bottleneck is the monitor, not the executor design.

**Limitations:**

yes

**Strengths And Weaknesses:**

## Strengths

1. **Good problem framing around efficiency, not just accuracy.** The paper’s most useful contribution may actually be the emphasis that medical LLM reasoning should be evaluated as an accuracy-cost tradeoff. Figure 1(a) is effective here, and it gives the paper a systems-oriented perspective that is relevant to ICML.
2. **Clear high-level architecture.** Figure 2 communicates the intended pipeline well. Even though I have reservations about the technical depth, the decomposition into monitoring, planning, and evaluating is easy to follow.
3. Adaptive routing seems empirically better than “always use everything.”** This is the most convincing empirical result. In Table 2, the comparison between MEDCOG-META and MEDCOG-ALL supports the authors’ thesis that external knowledge can hurt when applied indiscriminately.
4. **The paper includes interpretive analyses rather than only leaderboard numbers.** The threshold sweeps in Figure 4 and strategy distributions in Figure 3 are useful, even if I think some of the interpretations overreach.
## Weaknesses

1. **Routing equation inconsistency affecting reproducibility**
  The routing rule described in Equation (1) appears inconsistent with the textual explanation. The equation states that zero-shot reasoning is used when \((I_c \wedge I_f \wedge I_k = 0)\), which is satisfied whenever any indicator falls below threshold. However, the text states that zero-shot should be used only when all indicators are below threshold, and Section 3.3 claims SCoT is used when at least one indicator is positive. These definitions correspond to different routing policies and may change which reasoning strategy each sample receives.

2. **Limited methodological novelty behind the framework**
  The core mechanism is a prompted self-scoring module combined with threshold-based routing across existing strategies (zero-shot, SCoT, memory retrieval, and KG retrieval). Although the framework is presented as a meta-cognitive agent architecture, the actual implementation mainly relies on heuristic gating rather than a fundamentally new algorithmic component.

3. **Overstated claims regarding the “inference scaling law”**
  The paper suggests that it identifies and mitigates an inference scaling law, yet the evidence is limited to a logarithmic fit over a small Pareto frontier in Figure 1(b). While the reported \(R^2 = 0.91\) indicates correlation within the plotted points, this alone is insufficient to support the stronger claims made in the title and abstract.

- **Experimental evidence does not firmly establish consistent improvements**
  Several improvements reported on the hard subsets correspond to differences of only one or two questions, and the paper does not report confidence intervals or statistical significance testing. Additionally, some cost–accuracy comparisons rely on baseline numbers taken from prior work rather than rerunning all methods under the same evaluation harness, which weakens the central efficiency claim.

4. **Meta-cognitive interpretations may be conceptually overstated**
  The routing signals for complexity, familiarity, and knowledge density are produced through prompted self-assessment rather than demonstrated internal introspection. Interpreting these prompted scores as evidence of model “self-awareness,” “overconfidence,” or genuine meta-cognition therefore appears stronger than what the empirical setup directly supports.

---

> ### Author Rebuttal · Authors · 2026-03-31
>
> We thank Reviewer sXmp for the detailed and constructive feedback. We will revise manuscript as advised.
>
> >W1 & Q1: Equation (1).
>
> We thank the reviewer for catching the typo. The correct formulation is $I_c=0 \wedge I_f=0 \wedge I_k=0$ (zero-shot strategy is only triggered when all three scores are below thresholds). Our implementation and reported results strictly use this correct logic and we will revise the text.
>
> >W2: Novelty.
>
> Our core contribution is a **paradigm shift** from indiscriminate scaling of existing studies to instance-wise self-regulated reasoning, mitigating diminishing returns under scaling law.
>
> 1. **Meta-cognition Modeling**: Our meta-cognition-inspired architecture relies on LLM self-assessment for each query, rather than pre-defined heuristic rules. The meta-cognitive regulation and knowledge together form a meticulous architecture to validate the LLM's meta-cognitive regulation ability, where gating is one of the components.
>
> 2. **Calibrated Gating**: Raw LLM self-assessments often contain biases. Our calibration corrects biases and maps assessments to executor strategies. The assessments and calibrations also provide transparent insights for future studies.
>
> 3. **Approximation to Oracle**: While Oracle performance shows high potential and cannot be achieved through single or all strategies, our study provides a feasible pathway to approximate. We also quantify reasoning efficiency with density, IIE, and scaling curve.
>
> >W3: Claims regarding scaling law.
>
> We agree t o carefully calibrate our scope: Our aim is to mitigate the **diminishing returns** under scaling law to **maximize inference density**, not to solve the scaling law. Quantitative evidence is as follows:
>
> 1. Scaling Curve Stability: Inference density of MedCoG-Meta remains stably high with (a) 3 functions, (b) 5 hard sets, (c) random reference sets, demonstrating the robustness of our conclusions.
>
> |method|R^2|inference density|
> |-|-|-|
> |MedQA-H||||
> |log|0.91|5.46|
> |linear|0.80|2.65|
> |power law|0.92|57.13|
> |rand ref|0.84|5.77|
> |5 Hard sets|||
> |log|0.88|3.98|
>
> Full fitting results: https://anonymous.4open.science/r/MedCoG-anonymous-4555/curve_fitting.md
>
> 2. IIE metric: MedCoG-Meta achieves the highest IIE (0.443), a curve-agnostic metric, demonstrating our high efficiency.
>
> >W3 & Q2: Improvements under Statistical testing.
>
> We conduct paired t-tests with 95% confidence intervals (CI) across all 473 samples in Table 2.
>
> ||acc|delta acc|paired p-value|paired CI|
> |-|-|-|-|-|
> |AFlow|34.25|3.38|0.12|[-0.53%, 7.10%]|
> |MedCoG-SCoT|32.56|5.07|0.01|[1.66%, 8.79%]|
> |MedCoG-ALL|33.40|4.23|0.01|[1.06%, 7.61%]|
> |MedCoG-META|37.63||||
>
> - MedCoG-Meta significantly outperforms SCoT and ALL (p=0.01).
> - MedCoG-Meta achieves comparable performance as AFlow with only <50% of AFlow's cost, demonstrating superior efficiency.
>
> >W3 & Q3: Token & latency comparisons.
>
> Token/cost comparisons are harness-agnostic as cost strictly follows token counts (extracted from MedAgentsBench logs). Dollar cost is a reliable metric given different input/output pricing and latency varies across devices. We took MultiPersona, AFlow, MedAgents input/output logs from MedAgentsBench, where AFlow token is unavailable.
>
> **MedCoG-Meta achieves the lowest cost (25.8) and comparable latency while maintaining the highest accuracy.**
>
> |MedQA-H|input (k token)|output (k token)|total cost (1e-3 $)|total time (seconds)|
> |-|-|-|-|-|
> |MultiPersona|2.8|3.2|39.4|43.7|
> |AFlow|-|-|62.2|61.5|
> |MedAgents|20.9|4.7|98.8|58.2|
> |MedCoG-ALL|9.6|0.4|30.3|66.8|
> |MedCoG-Meta|7.2|0.6|25.8|62.7|
>
> Full cost table: https://anonymous.4open.science/r/MedCoG-anonymous-4555/cost.md
>
> >W4: Meta-cognitive interpretations.
>
> We agree that some statements are anthropomorphic.
> - Our architecture is architecturally inspired by meta-cognitive theory: the Monitoring/Planning/Evaluating pipeline mirrors meta-cognitive regulation, and knowledge taxonomy Procedural/Factual/Episodic reflects meta-cognitive frameworks. The prompted self-assessment is one component of this broader design.
> - Following prior work [MetaRAG, Reflexion, KnowSelf], we use anthropomorphic meta-cognitive phrases to describe analogous LLM behaviors and designs.
>
> We understand the potential misunderstandings and will revise the descriptions as "meta-cognition-inspired framework", "prompted self-assessment", and "overestimation".
>
> >Q4: Regulator accuracy.
>
> The accuracy and agreement of Regulator (52) to Oracle (67) is $52/67=77.61%$ on MedQA-H, while the regret from suboptimal routing is $1-77.61=22.39%$. We categorize the routing errors in **Table 6**.
>
> We agree that the bottleneck is the monitor, which is also our main focus. The 77.61% agreement with Oracle already yields significant efficiency and effective gains, showing the effectiveness of monitor. The gap further points out promising direction for future studies.
>
> The three score precision/recall/F1: https://anonymous.4open.science/r/MedCoG-anonymous-4555/regulator.md

---

> > ### Author Rebuttal · Reviewer_sXmp · 2026-04-02
> >
> > Author's rebuttal addresses my concerns.

---

> > > ### Author Response · Authors · 2026-04-03
> > >
> > > We thank the reviewer for the positive assessment and helpful feedback. We will incorporate the additional analyses and discussions in the final version of the manuscript.

---

### Official Review · Reviewer_BX5J · 2026-03-13

**Soundness:** 3
**Presentation:** 4
**Significance:** 3
**Originality:** 2
**Overall Recommendation:** 4
**Confidence:** 4

**Summary:**

This paper presents MedCoG, an agentic framework that deploys a set of reasoning strategies on-demand by self-assessing the complexity, familiarity, and knowledge density of a given task. The authors tackle the inference cost issue, proposing inference density and inference incremental efficiency as measures to quantify inference efficiency. Evaluated on five benchmarks with high difficulty, MedCoG outperforms existing methods and frameworks in terms of both accuracy and inference efficiency.

**Compliance With Llm Reviewing Policy:**

Affirmed.

**Final Justification:**

The rebuttal addressed my main concerns.

**Key Questions For Authors:**

1. Could the authors elaborate more on the inference setting behind "MedCoG-Oracle", which is stated as selecting the optimal strategy among the five reasoning strategies? A detailed explanation within the manuscript would be helpful for readability.

**Limitations:**

yes

**Strengths And Weaknesses:**

**Strengths**
1. The authors tackle an important issue that is often overlooked in existing medical agent studies: the excessive inference cost for maximum performance. This is logically supported by the inference cost/density analysis, placing the suggested method on a relatively ideal position between performance and efficiency.
2. The construction of MedCoG is meticulously done and shows deep consideration into various situations involved in LLM reasoning, such as mapping reasoning strategies to different meta-cognitive state vectors and categorizing different types of knowledge.
3. The performance-inference cost balance of MedCog is impressive, along with detailed analysis into LLMs' meta-cognitive characteristics.
4. The overall presentation of the conducted studies is clear and well-structured, improving readability.

**Weaknesses**
1. While the idea of selecting the optimal reasoning strategy per situation is original, MedCoG operates by sampling previously established methods, mainly Structural Chain-of-Thought (SCoT) attached with knowledge graphs and memory. This leads to concern into how original the method could be seen within the research community.
2.  In terms of utilizing episodic knowledge, the method requires a predefined database of past reasoning trajectories from similar tasks, which limits generalization and utility.
3. (Line 156) The statement that knowledge capacity and reasoning ability of LLMs are sufficient for medical reasoning seems to be an overstatement, as the oracle setting still involves knowledge graphs and past trajectories to reach high performance.

---

> ### Author Rebuttal · Authors · 2026-03-31
>
> We thank Reviewer BX5J for the constructive comments. We will revise the manuscript as advised.
>
> >W1: Originality: SCoT, KG, Memory are established.
>
> We sincerely thank the reviewer for pointing out our original idea of optimal reasoning strategy selection. Our core contribution is the **paradigm shift** from indiscriminately scaling to instance-wise self-regulated reasoning, rather than the invention of novel individual reasoning executors.
>
> 1. **Adaptive Reasoning Bottleneck**: While existing methods utilize CoT, KG, and memory, they fundamentally treat them as indiscriminately added computation. MedCoG introduced an LLM-centric on-demand adaptive regulation that improves accuracy with less noisy knowledge and reduces cost without indiscriminate scaling.
>
> 2. **Approximation to Oracle**: Adaptive regulation of strategies shows high Oracle performance. However, it is unreachable via any single strategy or all strategies of existing techniques. Such regulation mechanisms are not explored. Our study provides a feasible pathway to elicit this meta-cognitive regulation ability.
>
> 3. **Deliberate Experimental Validation Design**: It is a deliberate experimental design to use established executors (SCoT, KG, and Memory) in MedCoG framework to cover common knowledge types in medical reasoning studies (Appendix A.1 of Related Work), while modularizing the knowledge executors from regulations. This way, the contribution of meta-cognition regulation is isolated from executor design, making it possible to cleanly validate the regulation mechanism. On the contrary, if novel and complex knowledge executors are introduced, it would confound whether the gains are from regulation or the executors.
>
> 4. **Quantitative Efficiency Perspective**: Our originality also lies in a rethinking of existing strategies:
> - We provide a rethinking and Oracle analysis of existing strategies and identify the bottleneck as regulation rather than executors.
> - We quantify the inference efficiency problem to provide an efficiency perspective under scaling law.
>
> 5. **Extensibility**: MedCoG paradigm can easily adapt to novel reasoning techniques by adding powerful strategies to the strategy pool.
>
> >W2: Episodic knowledge generalization and utility.
>
> 1. Motivations: We design a memory mechanism as episodic knowledge to provide historical experiences to help reasoning. It is a knowledge type widely explored in medical reasoning studies [MedPrompt, MDTeamGPT, DoctorRAG, etc] and LLM Agent Memory studies.
>
> 2. Utility: The **utility is reasonable and controllable** for two reasons:
> - The growing open-sourced high-quality reasoning data in the community, such as MedReason, Huatuo-o1, etc. In our paper, the Case Bank is constructed from MedReason and does not require construction from scratch.
> - Construction from scratch is practical. We detailed our from-scratch construction pipeline and observed evolving effects in **Appendix I**. Thus, constructing a Case Bank catering to certain requirements and scenarios is implementable.
>
> 3. Generalization:
> - Our meta-cognitive regulation ensured generalization. When facing rare diseases or concepts, the familiarity score will be low, and the memory module will not be triggered, which avoids noise from episodic knowledge. It is supported by **decreased Memory Noise error samples in Table 6**.
> - The **generalization of Case Bank in out-of-distribution (OOD) settings is validated**:
>   - In **Table 2** and Sec 4.4 Effect of Case Bank Domain: MMLU/MMLU-Pro/PubMedQA cases are not included in the case bank. Their effectiveness demonstrates OOD generalization of Memory.
>   - In **Appendix G**: On PubMedQA, the OOD memory outperforms in-distribution memory, demonstrating OOD generalization and effectiveness of Memory.
>
> >W3: Line 156 statement.
>
> Thanks for pointing this out, and we agree completely. Our original intent was to state the high potential of Oracle to highlight the need for clever regulation in line 156. We will revise to "The high oracle performance indicates that the reasoning ability of LLMs with incorporated three knowledge types shows high potential for medical reasoning. "
>
> >Q1: MedCoG-Oracle setting.
>
> Thanks for your suggestions. MedCoG-Oracle is a post-hoc analytical setting for establishing the empirical upper bound of our strategy pool. In implementation, we independently execute all reasoning strategies for every sample in the dataset. The Oracle accuracy is evaluated under an "any-correct" condition: for a given question, if at least one of the five strategies is accurate, the Oracle accuracy is 1; if none of the strategies is accurate, the Oracle accuracy is 0. Oracle represents a perfect router that always knows which strategy would succeed. We will add the implementation details in lines 137-144.

---

> > ### Author Rebuttal · Reviewer_BX5J · 2026-04-04
> >
> > Thank you for the detailed response, my concerns have been adequately addressed.

---

> > > ### Author Response · Authors · 2026-04-04
> > >
> > > We thank the reviewer for the positive assessment and helpful feedback. All the discussions and clarifications will be fully included in the final version of the manuscript.

---

### Official Review · Reviewer_R35U · 2026-03-13

**Soundness:** 3
**Presentation:** 3
**Significance:** 3
**Originality:** 2
**Overall Recommendation:** 4
**Confidence:** 3

**Summary:**

This paper focuses on test-time scaling efficiency issues in medical reasoning, pointing out that more knowledge augmentation isn't always better, and proposes MedCoG, an on-demand reasoning framework driven by a meta-cognitive scheduler for dynamic selection. The authors also propose Inference Density and Inference Incremental Efficiency metrics to quantify reasoning efficiency and marginal gains. Experiments on multiple hard subsets of MedAgentsBench show that MEDCOG-META outperforms various baselines in both accuracy and IIE, and significantly outperforms MEDCOG-ALL, which always enables all knowledge.

**Compliance With Llm Reviewing Policy:**

Affirmed.

**Final Justification:**

The rebuttal has resolved my issue.

**Key Questions For Authors:**

- How much recalibration do meta-cognitive scores and threshold gating need across different backbones / prompt settings? Please provide cross-setting transfer or low-cost calibration results. If stability is better, I'd raise my assessment of practicality.
 - Does cost accounting fully include all steps like KG retrieval, entity alignment, ranking, and additional LLM calls? Please provide a more transparent cost breakdown table. If the cost definition is more rigorous, I'd have more confidence in `Inference Density/IIE` conclusions.
 - Inference Density depends on reference method sets and fitted curves. If reference sets or curve fitting methods are replaced, are conclusions stable? Maybe supplement the sensitivity analysis to enhance metric credibility.
 -  Current results are mainly on hard MC subsets. Do you have preliminary evidence that MedCoG is equally effective for open-ended medical reasoning and real multi-turn agent tasks? If so, it would significantly boost work impact.

**Limitations:**

yes

**Strengths And Weaknesses:**

Strengths
 - The paper explicitly models meta-cognitive scheduling as a calibratable strategy selection mechanism and proposes new evaluation perspectives around efficiency.
 - Test-time scaling costs in medical reasoning scenarios are high with diminishing marginal returns - a real problem. This work places performance-cost joint optimization at the center with clear practical significance.
 - Framework decomposition is clear, three knowledge types and threshold gating logic are easy to understand; error analysis and threshold studies supplement intuitive explanations.

Weaknesses
- Method relies heavily on meta-cognitive scores and threshold calibration. Stability across backbones, datasets, and prompts still needs more evidence. If frequent recalibration is needed, practical barriers would increase.
- Cost and accuracy partly depend on MedAgentsBench reports. Need clearer explanation of whether cost accounting includes all retrieval/ranking/additional call overhead, and whether conclusions are robust across different deployment environments.
- Inference Density metric depends on fitted reference curves and reference method sets. While authors supplement IIE for correction, metric interpretation may still mislead readers into thinking it's "absolute efficiency" rather than "relative reference frontier efficiency".
- Current empirical evidence mainly focuses on medical multiple-choice hard subsets; extrapolation to more open-ended clinical reasoning or real toolchain tasks hasn't been verified.

---

> ### Author Rebuttal · Authors · 2026-03-31
>
> We thank Reviewer R35U for the thoughtful and detailed feedback. We will revise manuscript as advised.
>
> >W1 & Q1: Stability of calibration.
>
> Our calibration is required across backbones and datasets. Each calibration requires monitor assessments of only 50-samples and a light weight parameter search over a strategy pool, which is fixed across prompts and LLMs.
>
> **MedCoG with low-cost median calibration shows stability in outperforming AFlow (48) across 5 datasets, 5 prompts, and 5 backbones, confirming stability.**
> - Datasets: GPT-4o with median calibration obtains $[52.0, 32.0, 30.1, 43.0, 15.0]$ (avg $34.6$) across 5 hard sets, outperforming MedCoG-ALL and AFlow.
> - Prompts: Accuracy of rephrased monitor prompts on MedQA-H is $[0.52, 0.53, 0.52, 0.50, 0.51]$ (avg $51.6$), outperforming MedCoG-ALL and AFLow with minimal variance.
> - Backbones: Median calibration yields comparable results, and most LLMs outperform AFLow, suggesting stability. Cross-setting suffers from meta-cognitive biases from GPT-4o, showing suboptimal results.
>
> ||GPT-4o|GPT-4o-mini|o3-mini|Gemini-2.0-Flash|Qwen3-8B|
> |-|-|-|-|-|-|
> |w/ calibration|52|50|49|52|47|
> |w/ GPT-4o threshold|52|40|44|48|46|
> |w/ median|52|48|49|47|42|
>
> We would like to clarify the necessity and practicality of calibrations:
> - Necessity: Calibration is necessary considering LLM meta-cognition behaviors, shown in **Figure 3&5**. With calibrating, the regulator obtains better precision and recall to trigger the correct strategy when needed, detailed in **pM5Z Q1**.
> - Practicality: Calibration cost in running 50 samples is low and controllable. Similar to agentic workflow optimization studies, calibrating to find optimal setting is reasonable. Moreover, thresholds are stable to perturbation in certain interval. For example, on MedQA-H, when the $\tau_f<=0.6$ and $\tau_k<=0.7$, accuracy is always above MedCoG-ALL.
>
> >W2 & Q2: Cost accounting.
>
> **Cost accounting fully includes all steps**. Per-sample breakdown (k-token and cost 1e-3$) is reported below.
>
> |MedQA-H|4o input|4o output|cost|4o-mini input|4o-mini output|cost|total cost|
> |-|-|-|-|-|-|-|-|
> |SCoT+KG|1.9|0.4|8.4|12.5|2.4|3.3|11.7|
> |SCoT+KG+Mem|9.6|0.4|28.5|6.5|1.3|1.7|30.3|
> |Meta Monitor|0.5|0.1|2.6||||2.6|
> |Meta Executor|6.7|0.4|21.1|7.8|1.5|2.1|23.2|
> |MedCoG-Meta|7.2|0.6|23.7|7.8|1.5|2.1|25.8|
>
> Full cost and KG function breakdown table: https://anonymous.4open.science/r/MedCoG-anonymous-4555/cost.md
>
> >W2: Stability across deployment environments
>
> Our conclusions are **stable across environments**.
> - Cost accounting is consistent across MedAgentsBench and our environments since most components rely solely on standardized API calls (price per token).
> - Conclusions are stable with various LLM series: GPT, Gemini, Qwen all show $>0.50$ precision/recall in meta-cognitive assessments (pM5Z Q1), demonstrating conclusion stability.
>
> >W3 & Q3: Scaling curve fitting stability and metric designs.
>
> 1. The conclusions are **stable with replaced reference sets (rand 2), 3 fitting methods (log/linear/power-law), and 5 hard sets**.
>
> |method|R^2|inference density|
> |-|-|-|
> |MedQA-H||||
> |log|0.91|5.46|
> |linear|0.80|2.65|
> |power law|0.92|57.13|
> |rand 2|0.84|5.77|
> |5 Hard sets|||
> |log|0.88|3.98|
> |power law|0.92|inf|
>
> Full fitting results: https://anonymous.4open.science/r/MedCoG-anonymous-4555/curve_fitting.md
>
> (1) Reference sets: We randomly mask 1/3 of the baselines. The inference density is between 3.25 and 17.67 for R^2>0.7 groups.
> - For reference, AFlow's density is 1.10 in rand 2 group (AFLow is not in reference set), less than 20% of our 5.77.
> - Pareto Frontier naturally filters low-accuracy/efficiency methods, representing current SOTA and ensuring robustness.
>
> (2) Fitting function:
> - Linear and power law functions also show high density ($>2$).
> - Linear groups show lower R^2 since the accuracy cannot linearly improve by adding cost.
> - Power law groups show comparable R^2, but density reaches infinity when accuracy (37.5) exceeds its upper bound (35.2), thus not perfect for scaling fitting.
> - Thus, we choose log function for stability and interpretability.
>
> (3) 5 hard sets: Inference density remains $>2$ across 5 datasets.
>
> 2. **Efficiency of MedCoG-Meta is fundamental and independent from fitting curve**: increasing accuracy via filtering noisy knowledge, and reducing cost via adaptive strategy.
>
> 3. We will polish inference density as relative frontier efficiency in the manuscript. The density highlights the efficiency gains above the scaling curve of frontier methods, which supports efficiency analysis.
>
> >W4 & Q4: Open-ended and multi-turn tasks.
>
> Results on MedQA-H-Open (LLM Judge 0-5) and AgentClinic (LLM Judge 0/1) demonstrate the **potential of MedCoG-Meta in open-ended and multi-turn tasks**.
>
> ||MedQA-H-Open(LLMScore>=4)|AgentClinic(LLMScore=1)|
> |-|-|-|
> |zero-shot|0.27|0.30|
> |SCoT|0.26|0.65|
> |SCoT+KG|0.31|0.65|
> |SCoT+Mem|0.28|0.70|
> |SCoT+KG+Mem|0.24|0.68|
> |Meta|0.34|0.73|

---

> > ### Author Rebuttal · Reviewer_R35U · 2026-04-04
> >
> > I have no other questions and have raised my score to 4. Good luck.

---

> > > ### Author Response · Authors · 2026-04-04
> > >
> > > We thank the reviewer for the positive assessment and constructive feedback. All the experiments and discussions provided during the rebuttal will be fully included in the final version of the manuscript.

---

### Decision · Program_Chairs · 2026-04-30

**Decision:**

Accept (regular)

**Comment:**

This submission tackles medical reasoning via LLMs especially on the perspective: improving the tradeoff between accuracy and cost at test time. Rather than merely scaling inference, it proposes MedCoG: a meta-cognitive routing framework that selects among reasoning strategies and knowledge resources relying on self-assessed metrics such as complexity, familiarity, and density.

Reviewers agree that framing of medical reasoning as a joint problem between performance and efficiency is effective. Several reviewers found that experimental findings that routing procedures better balances these two points than merely adding knowledge or reasoning steps. Various metrics formalized in the draft is useful, which has not been vastly explored.

On the other hand, reviewers pointed out routing mechanism remains heuristic, relying on prompted self-assessment and calibrated threshold. Stability and robustness should be further checked across different prompts and task settings. Some reviewers also concern that technical novelty exists mostly in system integration and routing than being fundamentally new. It was also questionable that which conclusions transfer to open-ended or real clinical settings. During author-reviewer discussions, the authors address majority of these concerns.

All in all, the submission is worthwile, contributing to LLM-based medical reasoning (especially through its focus on selective inference rather than mere scaling. The claimed meta-cognition brought empirical gains rather than working as a definite solutions to inference scaling or a strong demonstration of genuine model. We encourage the authors to further incorporate routing logic, assumptions, and additional analyses given in the rebuttal.